# Group-Aware Threshold Adaptation for Fair Classification

## Abstract

The fairness in machine learning is getting increasing attention, as its applications in different fields continue to expand and diversify. To mitigate the discriminated model behaviors between different demographic groups, we introduce a novel post-processing method to optimize over multiple fairness constraints through group-aware threshold adaptation. We propose to learn adaptive classification thresholds for each demographic group by optimizing the confusion matrix estimated from the probability distribution of a classification model output. As we only need an estimated probability distribution of model output instead of the classification model structure, our post-processing model can be applied to a wide range of classification models and improve fairness in a model-agnostic manner to ensure privacy. This even allows us to post-process existing fairness methods to further improve the trade-off between accuracy and fairness. Moreover, our model is efficient with low computational cost by alternating optimization and flexible with the optimization over multiple fairness constraints. We provide Pareto frontier to characterize fairness-accuracy trade-off. Also, we provide a theoretical analysis of the optimal thresholds obtained from our model in terms of both accuracy and fairness in classification. Experimental results demonstrate that our method outperforms state-of-the-art methods and obtains the result that is closest to the theoretical accuracy-fairness trade-off boundary.

## 1 Introduction

Machine learning is broadening its impact in various fields including autonomous driving, credit analysis, and job application screening. As a consequence, the role and importance of fairness in machine learning are emerging. However, recent models have been found to behave differently between demographic groups in favorable predictions. For example, it has been discovered that COMPAS, the criminal risk assessment software currently used to help pretrial release decisions, has biases between different races [4]. Specifically, blacks got higher risk scores predicted from the model than whites with similar profiles. Therefore, discrimination truly exists and resolving it in machine learning is very important and urgent because its direct and potential impact is growing tremendously.

However, obtaining fairness is not a trivial problem, because the data set itself will be biased when it is accumulated artificially. Simply removing or manipulating sensitive features (such as *race*, *gender*) from the data does not solve the bias, because there is indirect discrimination [19] or disparate treatment [1] due to the feature redundancy and relevance, which means sensitive information can be inferred from other features.

In order to alleviate discrimination from different perspectives, various quantitative measurements of group equity [7, 11, 2, 13] have been proposed. It has been proven that the pursuit of fairness is

subject to a trade-off between fairness and accuracy [14, 10], i.e., if we want to improve fairness, we need to sacrifice accuracy.

Moreover, Pleiss et al. [20] studied the trade-offs between fairness notions that cannot be satisfied at the same time. Therefore, recent works usually target at a certain fairness notion in different approaches such as pre-processing [6], in-processing [24], and post-processing [7] methods. However, these approaches suffer from the *lack of flexibility*, since it is difficult to adapt a fair model that is trained w.r.t. one certain fairness criterion so as to optimize over other fairness measures. If the fairness constraints change under some circumstances, traditional fairness models need to be re-trained from scratch, which is computationally demanding and sometimes inapplicable due to model settings. To overcome the limitations above, we propose a novel post-processing method to improve fairness in a model-agnostic manner. Our GSTAR (Group Specific Threshold Adaptation for faiR classification) model learns adaptive classification thresholds for each demographic group in classification task for improving the trade-off between fairness and accuracy. Given an existing classification model, GSTAR approximates the probability distribution of the model output via maximum likelihood estimation and utilizes confusion matrix to quantify accuracy and fairness w.r.t. the group-aware classification thresholds. This allows us to: 1) prevent from burdening additional complexity or deteriorate the stability of the training process of the classifier; 2) integrate different fairness notions into one unified objective function; 3) easily adapt one pre-trained model to other fairness constraints. We summarize our contributions of this paper as follows:

1. We propose a novel post-processing method, GSTAR, which can learn group-aware thresholds to optimize the trade-off between fairness and accuracy in classification. We derive rigorous theoretical analysis on the trade-off in our model, and empirically show that GSTAR outperforms state-of-the-art methods.

2. With GSTAR, we can simultaneously optimize over multiple fairness constraints with a low computational cost. GSTAR does not require multiple iterations over data, instead, it takes *at most* one pass of data in training for fast computation.

3. GSTAR can be adapted to a wide range of classification models in a model-agnostic manner and can adapt an existing classification model from one fairness criterion to another without re-training the classifier.

4. We derive Pareto frontiers of our model for the fairness-accuracy trade-offs that contextualize the quality of fair classification.

## 2   Related Works

In order to achieve group fairness, which quantifies the discrimination among different sensitive groups, a diverse notion of fairness has been introduced. Equalized odds [7] enforce equality of true positive rates and false positive rates between different demographic groups. Pleiss et al. [20] relaxed equalized odds to satisfy the calibration. Demographic parity or disparate impact [1] suggests that a model is unbiased if the model prediction is independent of the protected attribute.

Among different fairness methods, post-processing techniques propose to improve fairness by modifying the output of a black-box classifier. Hardt et al. [7] propose to ensure equalized odds by constraining the model output. Kamiran et al. [9] propose to give a favorable outcome to unprivileged and an unfavorable outcome to the privileged group when the confidence of the prediction is beyond a certain range. However, such *static* confidence window keeps the same regardless of the demographic group and is determined by grid search, so it is less efficient.

Threshold adjustment (a.k.a. thresholding) was introduced to improve the performance of *static* thresholds. In the literature, Menon et al. [18] prove that instance-dependent thresholding of the predictive probability function is the optimal classifier in cost-sensitive fairness measures. Also, when considering immediate utility, Corbett-Davies et al. [3] show that optimal algorithm is achieved from group-specific threshold which is determined by group statistics. However, to the best of our knowledge, the threshold adjustment approach has not been deeply studied that neither encompasses broad group fairness metrics nor describes an explicit method to achieve the threshold.

Trade-off between fairness and accuracy exists when we impose fairness constraint to a model. Recent studies [2, 25] prove that models targeting at such fairness notions conform to an information theoretic

lower bound on the joint error across different sensitive groups. Therefore, our work presents a practical upper bound of the best achievable accuracy given the fairness constraints.

Moreover, trade-offs between different fairness notions also exist if one has to consider multiple fairness criteria. Some of them are theoretically proven to be incompatible [6, 18, 14]. To express and formulate fairness, recent work [10] utilize confusion matrix and propose least-square accuracy-fairness optimization problem on multiple fairness notions, and categorize the trade-offs between the fairness notions.

Here, our work is the most related to the post-processing methods [7, 10]. Hardt et al. [7] propose a post-processing method that utilizes the mixing rate to meet the equalized odds. Ours is similar to Hardt et al. [7] in the manner that achieving group-wise threshold from the feasible region that is geometrically generated by the intersection between the receiver operating characteristic (ROC) curves conditioned on sensitive feature. Ours differ from [7] by generalizing the concept beyond equalized odds to other multiple fairness constraints into consideration. FACT [10] utilizes a single point (static) from the classifier to be post-processed as a reference which does not fully utilize the classifier for the post-processing. In contrast, by approximating the distribution of the continuous predicted logits, our GSTAR model enables a larger feasible region than [10] with a better fairness-accuracy trade-off. We validate the improvement in trade-off via both theoretical and empical results. It is notable that these related methods [7, 10] can be considered as a special case of GSTAR.

## 3 GSTAR for Fair Classification

### 3.1 Motivation

Consider a binary classification problem with a binary sensitive feature, such that the sensitive feature $A \in \{0, 1\}$ and label $Y \in \{0, 1\}$. In general, for a given data $X$, a binary classification model outputs an unnormalized logit $h(X) \in \mathbb{R}$ with the class label probability $p(X) = \sigma(h(X)) \in [0, 1]$, where $\sigma$ is an activation function (sigmoid function in logistic regression and neural network). It is not necessary to calculate $p$ in a classification model, e.g. support vector machines directly use the positiveness/negativeness of logit $h(X)$ to determine classification outcome. For traditional models, we use a cut-off threshold $\theta_h = 0$ for $h(X)$ (i.e., $\theta_p = \sigma(0) = 0.5$ for $p(X)$) in classification, such that the predicted label is determined by $\hat{Y} = \mathbb{I}\{h(X) \geq \theta_h\}$. In the following context, unless otherwise mentioned, we use $\theta$ to refer to the threshold $\theta_h$ on logit $h$ since it is applicable to a wider range of classification models, and the corresponding threshold on label probability $\theta_p$ can be easily inferred from the threshold on logit $h$. Traditional models use the same cut-off threshold $\theta$ for different demographic groups. However, since the distribution of logits $h$ in different demographic groups can be different, using the same threshold $\theta$ brings biased classification.

In Figure 1, we show a real-world example of image classification on CelebA dataset with ResNet50 [8] to show that the default setting of classification thresholds affects both accuracy and fairness in classification. The goal of this classification example is to predict the image of a person is whether attractive or not, and consider sensitive attribute as gender. This can be generalized to different sensitive attributes such as age or race [22, 16]. We can observe an obvious difference in the distribution of logit $h$ between two gender groups. In this case, if we use a unified classification threshold $\theta_1 = \theta_0 = 0$, it naturally brings a difference in the true positive rate and true negative rate between two gender groups, thus renders bias in classification. Instead, we observe that the optimal group-specific threshold obtained from GSTAR ($\theta_1^* > \theta_1$, and $\theta_0^* < \theta_0$) can adapt to such discrepancy in distribution between two demographic groups to improve both fairness and accuracy.

### 3.2 Group-Aware Classification Thresholds

Given an existing classification model and a sensitive attribute $a$, we can denote true positive rate (TP$_a$), false positive rate (FP$_a$), true negative rate (TN$_a$), and false negative rate (FN$_a$) in the confusion matrix. Most fairness notions can be represented with entries in the confusion matrix. For instance, Equal Opportunity (EOp) [7] requires $TP_0 = TP_1$, and Demographic Parity (DP) [1] requires

$$\frac{TP_1 n_{11} + FP_1 n_{01}}{N_1} = \frac{TP_0 n_{10} + FP_0 n_{00}}{N_0},$$

where $n_{ya}$ denotes the number of samples in the subset $\{Y = y, A = a\}$, $N_a = \sum_y n_{ya}$ denotes the number of samples in $\{Y = y\}$, and $N = \sum_{y,a} n_{ya}$ is the total number of samples.

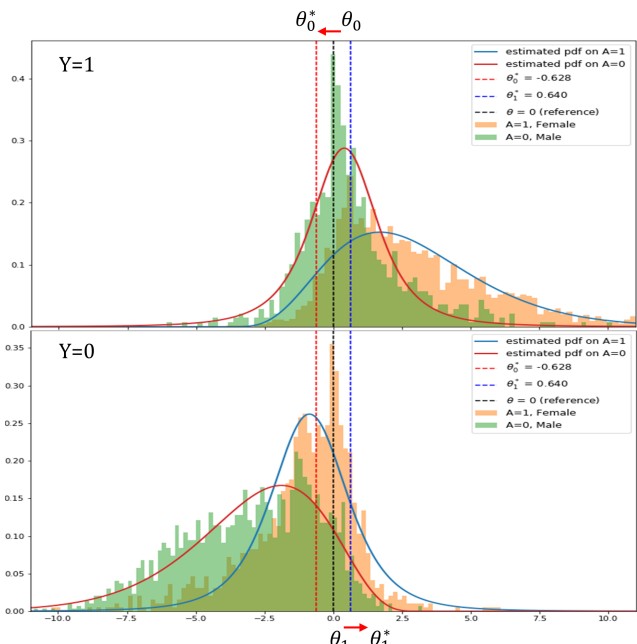

Figure 1: Histograms of logit $h$ distribution from logistic regression on CelebA data, where logit $h$ is used to determine the predicted label $\hat{Y} = \mathbb{I}\{h(X) \geq \theta\}$, and $\theta$ is the classification threshold. The top plot is for positive samples ($Y = 1$, attractive), and the bottom plot for negative samples ($Y = 0$, unattractive). In each plot, yellow/green bars represent the distribution of logit $h$ in different sensitive groups, and blue/red curves are estimated probability density functions of logit $h$ in different sensitive groups. $\theta_0 = \theta_1 = 0$ (black dashed line) are the default classification thresholds, that are identical for $A = 0$ and $A = 1$ groups. The default thresholds result in biased prediction towards the unprivileged group $A = 0$ due to the different logit $h$ distributions in different sensitive groups. $\theta_0^*$ (red dashed line), $\theta_1^*$ (blue dashed line) are group-aware thresholds from GSTAR for each sensitive group.

Consider the group-aware classification threshold $\boldsymbol{\theta} = (\theta_1, \theta_0)^\mathsf{T}$, where $\theta_a$ is the classification threshold for sensitive group $A = a$. We can formulate the entries in the confusion matrix w.r.t. $\boldsymbol{\theta}$ as below:

$$
\begin{aligned}
\text{TP}_a(\theta_a) &\approx 1 - \int_{-\infty}^{\theta_a} f_{1a}(x)dx, & \text{FN}_a(\theta_a) &\approx \int_{-\infty}^{\theta_a} f_{1a}(x)dx, \\
\text{FP}_a(\theta_a) &\approx 1 - \int_{-\infty}^{\theta_a} f_{0a}(x)dx, & \text{TN}_a(\theta_a) &\approx \int_{-\infty}^{\theta_a} f_{0a}(x)dx,
\end{aligned}
\tag{1}
$$

where $f_{ya}(x)$ is an estimated parametric probability density function of the distribution of output logit $h$ in the subset $\{Y = y, A = a\}$. Here, we consider gamma, Student's t, and normal distribution as the candidates for the estimated distribution, and select the one that has the maximum likelihood with the output distribution. Without loss of generality, this can be generalized with other parametric probability density function based on the needs or prior knowledge.

Then, we formulate the fairness-constrained classification problem with the objective of minimizing classification error into a least-squared optimization problem. We denote our objective function as $\mathcal{L}(\boldsymbol{\theta})$ which consists of the performance loss $\mathcal{L}_{per}(\boldsymbol{\theta})$ and fairness loss $\mathcal{L}_{fair}(\boldsymbol{\theta})$. $\mathcal{L}_{per}(\boldsymbol{\theta})$ and $\mathcal{L}_{fair}(\boldsymbol{\theta})$ measures the error in performance and fairness respectively that are represented with the entries of the confusion matrix. In other words, our goal is to minimize the objective function $\mathcal{L}(\boldsymbol{\theta})$ as below:

$$
\mathcal{L}(\boldsymbol{\theta}) = \mathcal{L}_{per}(\boldsymbol{\theta}) + \lambda \mathcal{L}_{fair}(\boldsymbol{\theta}),
\tag{2}
$$

where $\lambda$ is a hyperparameter that determines how much fairness is enforced in the optimization.

The performance error $\mathcal{L}_{per}(\boldsymbol{\theta})$ can be written as

$$
\mathcal{L}_{per}(\boldsymbol{\theta}) = \left( \frac{n_{01}}{N}\text{FP}_1(\theta_1) + \frac{n_{11}}{N}\text{FN}_1(\theta_1) + \frac{n_{00}}{N}\text{FP}_0(\theta_0) + \frac{n_{10}}{N}\text{FN}_0(\theta_0) \right)^2.
\tag{3}
$$

As for $\mathcal{L}_{fair}(\boldsymbol{\theta})$, it can be formulated to any fairness metrics that are expressible with confusion matrix. For instance, when we impose EOp ($TP_1 = TP_0$) and predictive equality (PE) ($FP_1 = FP_0$) [2], we can get the corresponding $\mathcal{L}_{fair}(\boldsymbol{\theta})$ by summing over the least squared form of each constraint. Also, satisfying EOp and PP is equivalent to satisfying Equalized Odds (EOd) [7], This can be formulated in our $\mathcal{L}_{fair}$ as

$$
\begin{aligned}
\mathcal{L}_{fair}^{EOd}(\boldsymbol{\theta}) = & \mathcal{L}_{fair}^{EOp}(\boldsymbol{\theta}) + \mathcal{L}_{fair}^{PP}(\boldsymbol{\theta}) \\
= & \left(TP_1(\theta_1) - TP_0(\theta_0)\right)^2 + \left(FP_1(\theta_1) - FP_0(\theta_0)\right)^2.
\end{aligned}
\tag{4}
$$

Note that a lower $\mathcal{L}_{fair}$ value indicates a fairer threshold. When $\mathcal{L}_{fair}^{EOD}(\boldsymbol{\theta}) = 0$, we can interpret as the $\boldsymbol{\theta}$ satisfies the perfect EOd fairness.

Similar to (4), we can enforce multiple fairness constraints by summing over the least squared of each metric with different weight constant $\lambda$ to each fairness constraints if needed.

Also, it is notable that compared to the recent paper [10] that enforces fairness through confusion tensor, our formulation of fairness in $\mathcal{L}_{fair}(\boldsymbol{\theta})$ represents a direct notion of fairness metrics and improves the measures that allows us to achieve better performance and Pareto frontiers that is shown in Section 4.2 and Figure 2. For example, $\boldsymbol{A}_{\text{EOd}}$ in the paper is calculated as $M_1 \text{EOp} + M_0 \text{PE}$, where $M_y = n_{y0} + n_{y1}$, such that EOd is a weighted sum of EOp and PE with weights being the number of samples in each class. In this expression, the imbalance between the two fairness criteria will grow as the degree of imbalance in the data increases. In contrast, our formulation expresses the constraints as the exact notion of each metric that is not biased by the statistics of the datset and we observe improved Pareto frontier as in Figure 2.

We propose to optimize our threshold $\boldsymbol{\theta}$ with alternating optimization method. Here we take EOp constraint as an example to show the alternating optimization steps, then $\mathcal{L}_{fair}(\boldsymbol{\theta})$ can be written as

$$
\mathcal{L}_{fair}^{EOp}(\boldsymbol{\theta}) = \left(TP_1(\theta_1) - TP_0(\theta_0)\right)^2.
\tag{5}
$$

**The first step** is to fix $\theta_0$ and update $\theta_1$. We can approximate the terms that are related to $\theta_1$ (e.g., $TP_1, FP_1, TN_1, FN_1$) in (1) with first-order Taylor expansion at $\theta_1^{\tau-1}$. For example,

$$
TP_1(\theta_1) \approx TP_1(\theta_1^{\tau-1}) + \left.\frac{\partial TP_1}{\partial \theta_1}\right|_{\theta_1 = \theta_1^{\tau-1}} (\theta_1 - \theta_1^{\tau-1})
\tag{6}
$$

From (1), we can easily derive that

$$
\begin{aligned}
TP_1(\theta_1^{\tau-1}) = & 1 - \int_{-\infty}^{\theta_1^{\tau-1}} f_{11}(x) dx, \\
\frac{\partial TP_1}{\partial \theta_1} = & -f_{11}(\theta_1^{\tau-1}).
\end{aligned}
\tag{7}
$$

Similarly, we can find the first order Taylor expansion of $FP_1, FN_1$, and $TN_1$. Then, the update of $\theta_1$ w.r.t. (2) can be approximated with the following minimization problem w.r.t. $\Delta_1$

$$
\Delta_1^{\tau} := \underset{\Delta_1}{\arg\min} (\eta^{\tau} + \alpha^{\tau} \Delta_1)^2 + \lambda (\epsilon^{\tau} + \beta^{\tau} \Delta_1)^2,
\tag{8}
$$

where $\Delta_1 = \theta_1 - \theta_1^{\tau-1}$ and

$$
\begin{aligned}
\alpha_1^{\tau} = & \frac{n_{11}}{N} f_{11}(\theta_1^{\tau-1}) - \frac{n_{01}}{N} f_{01}(\theta_1^{\tau-1}), \\
\beta_1^{\tau} = & -f_{11}(\theta_1^{\tau-1}), \\
\eta_1^{\tau} = & \int_{-\infty}^{\theta_1^{\tau-1}} \left(\frac{n_{11}}{N} f_{11}(x) + \frac{n_{01}}{N}(1 - f_{01}(x))\right) dx + \int_{-\infty}^{\theta_0^{\tau-1}} \left(\frac{n_{10}}{N} f_{10}(x) + \frac{n_{00}}{N}(1 - f_{00}(x))\right) dx, \\
\epsilon_1^{\tau} = & \int_{\infty}^{\theta_1^{\tau-1}} f_{11}(x) dx - \int_{\infty}^{\theta_0^{\tau-1}} f_{01}(x) dx.
\end{aligned}
\tag{9}
$$

Taking the derivative of (8) w.r.t. $\Delta_1$ and setting it to 0, we can easily obtain the closed-form solution of $\Delta_1^{\tau}$ as

$$
\Delta_1^{\tau} = -\frac{\alpha^{\tau} \eta^{\tau} + \lambda \beta^{\tau} \epsilon^{\tau}}{(\alpha^{\tau})^2 + \lambda (\beta^{\tau})^2}.
\tag{10}
$$

---
**Algorithm 1** Optimization Algorithm of GSTAR Model
---

**Input** dataset $\mathcal{X} \times \mathcal{A} \times \mathcal{Y} = \{(\mathbf{x}_i, \mathbf{a}_i, \mathbf{y}_i)\}_{i=1}^n$, classification model $h(X)$, hyperparameter $\lambda$.
**Output** Group-specific threshold $\boldsymbol{\theta} = (\theta_1, \theta_0)$.
**Initialize** $\boldsymbol{\theta} = (\theta_1, \theta_0) = (0, 0)$.
1. Given a classifier $H(x)$, estimate probability density function $f_{ya}, y, a \in \{0, 1\}$ by maximum likelihood estimation.
**while** not converge **do**

    2. Calculate the optimal step $\Delta_1$ as $\Delta_1 = -\frac{\alpha_1 \eta_1 + \lambda \beta_1 \epsilon_1}{\alpha_1^2 + \lambda \beta_1^2}$, with $\alpha_1, \beta_1, \eta_1, \epsilon_1$ values shown in (9);

    3. Update the threshold: $\theta_1 \leftarrow \theta_1 + \Delta_1$;

    4. Calculate the optimal step $\Delta_0$ as $\Delta_0 = -\frac{\alpha_0 \eta_0 + \lambda \beta_0 \epsilon_0}{\alpha_0^2 + \lambda \beta_0^2}$ with $\alpha_0, \beta_0, \eta_0, \epsilon_0$ values calculated in a similar way as in (9):

    5. Update the threshold: $\theta_0 \leftarrow \theta_0 + \Delta_0$.
**end while**

---

**The second step** is to fix $\theta_1$ and update $\theta_0$, and this can be achieved in a similar way of updating $\theta_1$. Then we can finalize the alternating optimization as:

$$\begin{aligned} \theta_0^\tau &= \theta_0^{\tau-1} + \Delta_0^\tau, \\ \theta_1^\tau &= \theta_1^{\tau-1} + \Delta_1^\tau. \end{aligned} \tag{11}$$

It is notable that in each iteration we derive the optimal update step $\theta$, which eliminates the burden of tuning hyperparameter (such as learning rate) in iterative algorithm. The optimization step is summarized in Algorithm 1. The above algorithm can easily extend to multiple fairness constraints by adding corresponding squared-loss fairness terms to (2).

**Time Complexity:** The alternating optimization of GSTAR model is of low computational cost. We take at most one pass of the data for learning the estimated probability density functions $f_{ya}$ in (1) (we do not even need to traverse the data if the parameters (such mean and variance in Gaussian distribution) for the estimated probability density functions $f_{ya}$ can be provided). The optimization of $\boldsymbol{\theta}$ with alternating optimization is efficient since we only need $f_{ya}$ as we have seen in (9) and (10). $\boldsymbol{\theta} \in \mathbb{R}^2$ is a vector with fixed small size. Therefore, we need a constant time for each update. Overall, the time complexity of GSTAR is $O(n + T)$, where $n$ is the number of samples, and $T$ is the number of iterations in alternating optimization.

We further derive the theoretical analysis of our GSTAR model on the balance between fairness and accuracy, which indicates that the optimal solution provides guarantees on model accuracy under the optimal fairness constraint. Details of the theoretical analysis is in the Supplementary material. Besides, if a unified threshold is necessary [3], i.e., $\theta_1 = \theta_0$, the optimization algorithm also applies and we only have one scalar variable in (2). When we have a unified threshold, we do not require sensitive information in the testing phase that we can conform more strict privacy regulations than group-aware thresholding. However, we have to sacrifice both fairness and accuracy as the thresholding is less flexible.

## 4 Experiments

In this section, we validate GSTAR model on four well-known fairness datasets and compare with other state-of-the-art methods. First, we plot Pareto frontiers of ours and FACT (MS) [10] to demonstrate the trade-offs between fairness and accuracy. Second, we evaluate the models with different fairness metrics and validate that our model is highly adaptive to any fairness metrics that are expressible with confusion matrix [7, 11, 2, 1]. Third, we use our model as a post-processing method to existing fair models and show that our model further improves existing fair models in an efficient and model-agnostic manner.

### 4.1 Experimental Setup

We compare with multiple fairness approaches in the experiments. For clear demonstration of results, we use different shapes of marker for each comparing methods in Figure 2 and Figure 4. The comparing methods include: **Learning fair representations for kernel models** (abbreviated as FGP) [23],

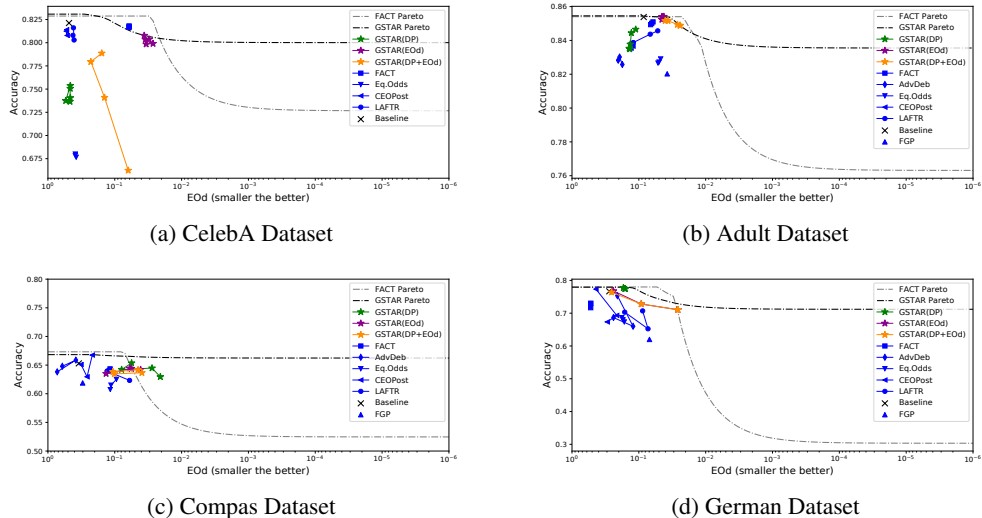

(a) CelebA Dataset

(b) Adult Dataset

(c) Compas Dataset

(d) German Dataset

Figure 2: Model-specific Pareto frontiers of equalized odds to show the upper bound of best achievable accuracy under different fairness constraints. Upper right region under the boundary shows better fairness and higher accuracy. We plot three variations of GSTAR (star-shaped) with different fairness objectives. GSTAR is the closest to the Pareto frontier which indicates the best trade-offs.

**Fairness confusion tensor** (abbreviated as FACT) [10], **Disparate impact remover** (abbreviated as DIR) [6], **Adversarial de-biasing** (abbreviated as AdvDeb) [24], **Calibrated equalized odds post-processing** (abbreviated as CEOPost) [20], **Equality of opportunity in supervised learning** (abbreviated as Odds) [7], **Learning adversarially fair and transferable representations** (abbreviated as LAFTR) [17], and **Baseline**: For CelebA dataset, we use ResNet50 [8] as a reference, and logistic regression for all other datasets. Our method is optimized with $\lambda$ in the range of $[10^{-1}, 10^4]$ with alternating optimization method. All experiments are implemented with Pytorch framework on i9-9960X CPU and a Quadro RTX 6000 GPU.

We choose broadly used fairness metrics in evaluation including: **equal opportunity difference** and **equalized odds difference** (abbreviated as EOp, and EOd respectively) [7] ; **1-disparate impact** (abbreviated as 1-DIMP) [1]; **balanced accuracy difference** (abbreviated as BD).

We evaluate the methods on four fairness datasets: **CelebA** image dataset[1] [15], **Adult** dataset from the UCI repository [12], **COMPAS**[2] (Correctional Offender Management Profiling for Alternative Sanctions) dataset, and **German** credit dataset from the UCI repository [5]. All data is split as 70% for training and 30% for testing. More details of the comparing methods, evaluation metrics, and datasets are provided in the Supplementary material.

## 4.2 Performance and Fairness-Accuracy Trade-Offs

In this subsection, we look into the performance evaluation of GSTAR comparing with other state-of-the-art methods. We consider Pareto frontier to visualize the trade-offs between fairness and accuracy to demonstrate the measure of performance.

In Figure 2, we plot Pareto frontier, which is the upper bound for the accuracy-fairness trade-offs, desired output locates at the upper right region under the boundary which corresponds to higher values in accuracy and lower values in fairness discrepancy. With the same fairness constraints are given, we achieve a better frontier than the FACT [10] as we equally weigh on demographic statistics and have a better feasible region. To obtain our results (star points), we first estimate the logit distribution from the output of the baseline model, and then we get optimal adaptive thresholds with corresponding fairness metric by updating w.r.t. the objective function in (2). Here we have three combinations of fairness imposed to GSTAR: demographic parity (DP), equalized odds (EOd), and

---

[1]http://mmlab.ie.cuhk.edu.hk/projects/CelebA.html
[2]https://github.com/propublica/compas-analysis

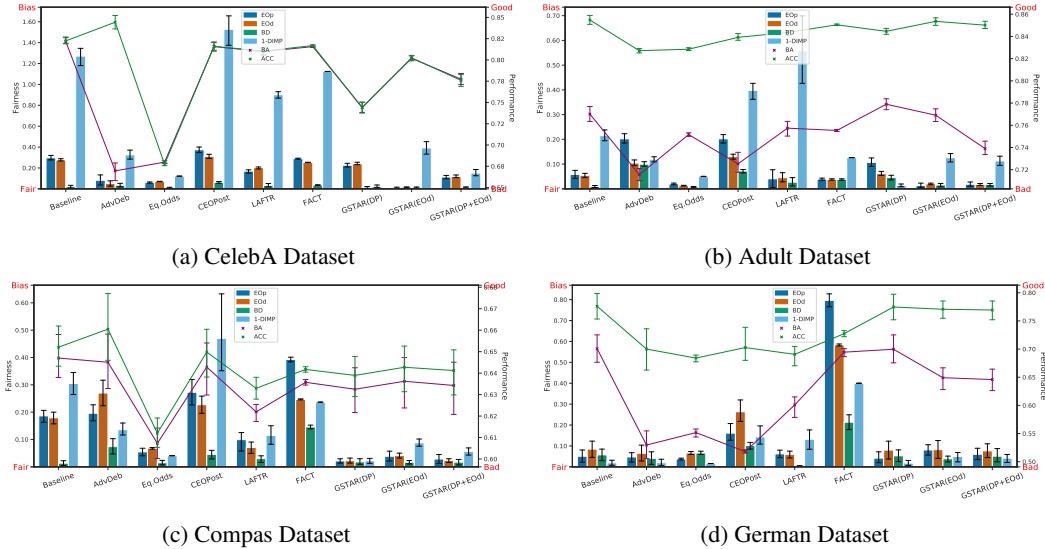

(a) CelebA Dataset

(b) Adult Dataset

(c) Compas Dataset

(d) German Dataset

Figure 3: Quantitative evaluation on fairness and performance metrics. The bar plots indicate fairness measures (EOp, EOd, BD, 1-DISP) of each model. Lower fairness values in the left y-axis shows better fairness. The line plots indicate the performance measure (balanced accuracy (BA) and accuracy (ACC)) of each model. Higher performance values in the right y-axis shows better classifcation performance. We consider three variations of GSTAR models (DP, EOd, DP+EOd).

with both constraints (DP+EOd). By post-processing on a simple baseline, we achieved significantly better fairness with small or no sacrifice in accuracy. In all datasets, GSATR got competitive or better results than other state-of-the-art methods on both fairness and accuracy.

For example, we got $\boldsymbol{\theta}^*_{EOd} = (0.640, -0.627)^\mathsf{T}$ for the CelebA dataset. This shows that we have a higher threshold for the privileged group and a lower threshold for the unprivileged group. This optimal thresholding from GSTAR allows more samples from the privileged group to be correctly predicted as unattractive that would compensate for the discrimination of the original model. In other words, this improves predictive equality [2] with a huge amount from 0.235 to 0.014. Also, true positive rate difference (also known as equality of opportunity [7]) got reduced from 0.282 to 0.018. It is notable that GSTAR only sacrificed 2.2% of accuracy to bring the big improvement in fairness.

Since the objective function of our model is independent to data dimensionality, our model is much more efficient especially for high dimensional data. We mostly outperform the computational cost comparing to the other methods. The comparison of computational time on the datasets can be found in the Supplementary material.

## 4.3 Flexibility and Multiple Fairness Constraints

Since each fairness metric has different interests, it has been theoretically proven that they cannot be perfectly satisfied all together [20, 2, 11]. Because of this inherent trade-offs between fairness metrics, most of the recent works focus on a single metric at a time to achieve fairness. However with GSTAR, we have the flexibility to optimize on multiple fairness constraints that can be represented in the confusion matrix format. Moreover, given the estimated distribution $f_{ya}$ of a black-box classification model, we can adjust the optimal $\boldsymbol{\theta}$ based on the needs by accommodating different fairness criteria.

Figure 3 demonstrates the result of the methods with fairness metrics and accuracy trade-off evaluations. Overall, the variations of GSTAR achieve the best fairness on each target fairness while preserving the performance. For example in Figure 3(a), GSTAR with EOd constraint has outstanding performance in most fairness metrics with comparable accuracy (80.3%). Comparing with GSTAR (EOd), when we introduce EOd and DP together (DP+EOd), we achieve significantly better w.r.t. DP fairness with sacrificing a small amount of accuracy and EOd.

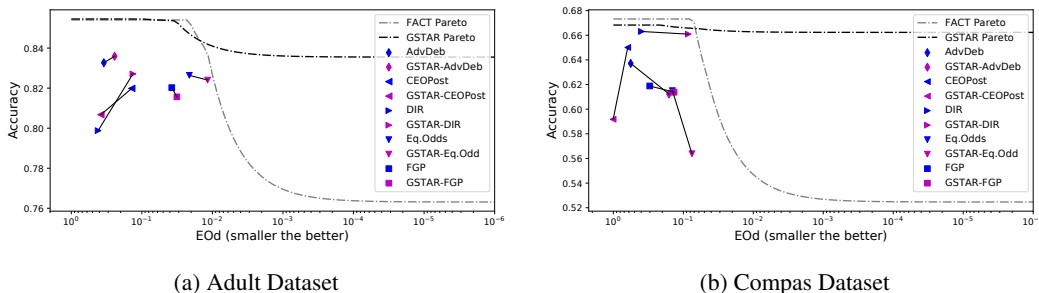

(a) Adult Dataset       (b) Compas Dataset

Figure 4: Illustration of post-processing (magenta colored points) on existing fairness models (blue colored points). Given the outputs of each model, we efficiently improve existing fairness models with optimized group-aware thresholds from GSTAR.

In general, by sacrificing individual fairness performance, we could introduce multiple constraints. Also, we implicitly observe that the more fairness constraints are introduced, the more accuracy is sacrificed. We empirically found that in some cases (e.g. Figure 3(c)), introducing multiple fairness is complementary to each other that improves both conditions.

### 4.4 Post-Processing on an Existing Fair Model

For a binary classifier that has a single fixed classification threshold (0 for out logit, and 0.5 for label probability), we can improve the trade-off between fairness and accuracy via GSTAR post-processing. Given the logit/probability of the dataset from a black-box model, we can improve the fairness as illustrated in Figure 4. In most cases, we observe improvement in fairness after GSTAR post-processing. It is also interesting to note that by optimizing the different thresholds for each protected group, we even obtain better performance on both fairness and accuracy, which indicates that the threshold optimization can not only improve fairness but also accuracy.

However, when the distribution of the logits/probability is highly extreme (such as the results of using GSTAR to post-process CEOPost), it is difficult to estimate the distribution and thus causes erroneous optimization in GSTAR. We empirically found that when the dataset is extremely imbalanced such that we do not have enough samples to estimate the logit/probability distribution, or black-box model is too certain to the prediction that samples are concentrated to certain output, this problem arises.

## 5 Conclusion and Discussion

In this paper, we propose a group-aware threshold adaptation method (GSTAR) to post-process a black-box model and optimize over multiple fairness constraints.We directly optimize the classification threshold for each demographic group w.r.t. the classification error and multiple fairness constraints in a unified objective function, such that we can practically achieve an optimal trade-off between accuracy and fairness in fair classification. Our method is applicable to diverse notions of group fairness as the majority of fairness notions can be expressed as a linear or quadratic equation through confusion matrix. We empirically show that GSTAR is *flexible* with fairness regularization, *efficient* with low computational cost. We also notice that the adaptive thresholds benefit accuracy in some cases. GSTAR agrees to protect *privacy* such as article 17 of EU's GDPR [21] with model-agnostic post-processing. We only require the estimated distribution of the output from a black-box model i.e., our post-processing method is oblivious to features. Thus training data is no longer needed and allowed to be discarded after training the black-box model.

Further, we empirically find that GSTAR is not applicable to post-process some classification models in the following situations: 1) the model does not provide logit/probability as the outcome; 2) The model provides an extreme distribution of the output logit/probability. For example, when the model is too certain about its prediction, it will be difficult to perform probability density estimation. In our future work, we will study possible strategies to solve the above limitations, and extend GSTAR to multi-class, multi-sensitive group problems and improve the fairness-accuracy trade-off in a more general scheme.

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
