# Supplementary Material for "Group-Aware Threshold Adaptation for Fair Classification"

## 1 Upper Bounds on False-Positive/Negative Rate Gap Between Groups

### 1.1 Notations

We start from defining notations. We denote $f_{ya}(x)$ for the estimated parametric probability density function (PDF) of the distribution of output logit $h$ in the subset $\{Y = y, A = a\}$. Correspondingly, we denote the corresponding cumulative distribution function (CDF) as

$$F_{ya}(x) = \int_{-\infty}^{x} f_{ya}(x)dx.$$

We use $F_{ya}^{-1}(x)$ to denote the inverse of the CDF.

Then, following the definitions given in the main paper, we have

$$
\begin{aligned}
\text{TP}_a(\theta_a) &= 1 - F_{1a}(\theta_a), & \text{FN}_a(\theta_a) &= F_{1a}(\theta_a), \\
\text{FP}_a(\theta_a) &= 1 - F_{0a}(\theta_a), & \text{TN}_a(\theta_a) &= F_{0a}(\theta_a).
\end{aligned}
\tag{1}
$$

### 1.2 Characterizing the accuracy loss function under perfect EOp condition

Before stating the theorem, we illustrate the difference between $\mathcal{L}_{per}(\boldsymbol{\theta})$ used in our paper versus loss function one would use in a population-wise classification problem (without considering group-aware thresholds). That is, one would only consider the loss function on accuracy

$$\bar{\mathcal{L}}_{per}(\theta) = \left(r_1 \bar{\text{FN}}(\theta) + r_0 \bar{\text{FP}}(\theta)\right)^2, \tag{2}$$

where only one threshold $\theta$ (for both groups) needs to be decided, $r_y = (n_{y0} + n_{y1})/N$ is the population ratio of data samples with label $y$, $\bar{\text{FN}}(\theta), \bar{\text{FP}}(\theta)$ are the population-wise false-negative and false-positive rate. $\bar{\text{FN}}(\theta), \bar{\text{FP}}(\theta)$ are defined in a similar way as in (1) except that we just use the population-wise pdf $\bar{f}_y(x)$ in the integral for label $y$. (2) will be our benchmark to compare with $\mathcal{L}_{per}(\boldsymbol{\theta})$ used in our paper.

We start from considering the case that we achieve perfect EOp condition, that is

$$\text{TP}_1(\theta_1) = \text{TP}_0(\theta_0), \quad \text{or equivalently} \quad \text{FN}_1(\theta_1) = \text{FN}_0(\theta_0). \tag{3}$$

This means that $\theta_0$ and $\theta_1$ satisfies the following condition

$$F_{11}(\theta_1) = F_{10}(\theta_0). \tag{4}$$

Equivalently, we have

$$\theta_0 = F_{10}^{-1}\left(F_{11}(\theta_1)\right). \tag{5}$$

Submitted to 35th Conference on Neural Information Processing Systems (NeurIPS 2021). Do not distribute.

Under any given pair of $(\theta_0, \theta_1)$ that satisfies (5), recall that the performance error $\mathcal{L}_{per}(\boldsymbol{\theta})$ is defined as

$$\mathcal{L}_{per}(\boldsymbol{\theta}) = \left(\frac{n_{01}}{N}\text{FP}_1(\theta_1) + \frac{n_{11}}{N}\text{FN}_1(\theta_1) + \frac{n_{00}}{N}\text{FP}_0(\theta_0) + \frac{n_{10}}{N}\text{FN}_0(\theta_0)\right)^2. \tag{6}$$

From (3), we get

$$\frac{n_{11}}{N}\text{FN}_1(\theta_1) + \frac{n_{10}}{N}\text{FN}_0(\theta_0) = \frac{n_{11} + n_{10}}{N}\text{FN}(\theta_1) = r_1\text{FN}(\theta_1),$$

where $r_1$ denotes, over the entire population (across different groups), proportion of samples with positive labels. In other words, $r_1\text{FN}(\theta_1)$ represents the proportion of data samples (from both groups) with positive label but falsely classified as negative out of the entire dataset.

Next, we look at the other two terms:

$$\frac{n_{01}}{N}\text{FP}_1(\theta_1) + \frac{n_{00}}{N}\text{FP}_0(\theta_0).$$

This sum can be written as

$$\begin{aligned}
\frac{n_{01}}{N}\text{FP}_1(\theta_1) + \frac{n_{00}}{N}\text{FP}_0(\theta_0) &= \frac{n_{01} + n_{00}}{N}\text{FP}_1(\theta_1) + \frac{n_{00}}{N}\left(\text{FP}_0(\theta_0) - \text{FP}_1(\theta_1)\right) \\
&= r_0\text{FP}(\theta_1) + \frac{n_{00}}{N}\left(\text{FP}_0(\theta_0) - \text{FP}_1(\theta_1)\right).
\end{aligned}$$

We denote $\epsilon_1 = \left(\text{FP}_0(\theta_0) - \text{FP}_1(\theta_1)\right)$. Hence,

$$\mathcal{L}_{per}(\boldsymbol{\theta}) = \mathcal{L}_{per}(\theta_1) = \left(r_1\text{FN}(\theta_1) + r_0\text{FP}(\theta_1) + \frac{n_{00}}{N}\epsilon_1\right)^2. \tag{7}$$

Comparing (2) with (7), we can see that, when $\text{FP}_0(\theta_0) > \text{FP}_1(\theta_1)$, the term $\frac{n_{00}}{N}\epsilon_1$ captures the additional accuracy loss due to that we have chosen two different thresholds even though that condition (4) is satisfied. Next, we characterize an upper bound for $\epsilon_1$.

### 1.3 Theorem 1 and its Proof

We first state the assumptions we need to make for Theorem 1.

**Assumption 1.** *For any given classier $h$ and its induced parametric PDF $f_{ya}$ and CDF $F_{ya}$, we assume the following holds:*

- *The PDF $f_{ya}(x)$ is uniformly bounded, i.e., there is an $\hat{f}_{ya}(x) = \max_x f_{ya}(x)$.*

- *The inverse CDF $F_{ya}^{-1}(x)$ is Lipschitz continuous with Lipschitz constant $M_{ya}$.*

- *The difference in the CDF between two groups is uniformly bounded, i.e.,*

$$|F_{y1}(x) - F_{y0}(x)| \le u_y, \ \forall x.$$

**Theorem 1.** *For any given classifier that satisfies Assumption 1 and any given pair of thresholds $(\theta_0, \theta_1)$ that satisfies the perfect EOp condition, the gap between false-positive rates of the two group is upper bounded by*

$$|\epsilon_1| = \left|FP_0(\theta_0) - FP_1(\theta_1)\right| \le u_0 + C_1 u_1, \tag{8}$$

*where $C_1 = \hat{f}_{01}M_{10}$.*

*Proof.* Recall that $\text{FP}_1(\theta_1) = 1 - F_{01}(\theta_1)$ and $\text{FP}_0(\theta_0) = 1 - F_{00}(\theta_0)$. Hence,

$$\begin{aligned}
\left|\text{FP}_0(\theta_0) - \text{FP}_1(\theta_1)\right| &= \left|F_{01}(\theta_1) - F_{00}(\theta_0)\right| \\
&\le \left|F_{01}(\theta_1) - F_{01}(\theta_0)\right| + \left|F_{01}(\theta_0) - F_{00}(\theta_0)\right|.
\end{aligned}$$

To bound $\epsilon$, we just need to bound $\left|F_{01}(\theta_1) - F_{01}(\theta_0)\right|$ and $\left|F_{01}(\theta_0) - F_{00}(\theta_0)\right|$.

For the second one, we note that from Assumption 1 that

$$\left|F_{01}(\theta_0) - F_{00}(\theta_0)\right| \le u_0.$$

44 For the first one, we note that

$$\left| F_{01}(\theta_1) - F_{01}(\theta_0) \right| \leq \hat{f}_{01} |\theta_1 - \theta_0|,$$

45 where $\hat{f}_{01} = \max_x f_{01}(x)$.

46 Next, we bound $|\theta_1 - \theta_0|$. Note that from (5),

$$
\begin{aligned}
|\theta_1 - \theta_0| &= \left| F_{10}^{-1}\big(F_{11}(\theta_1)\big) - \theta_1 \right| \\
&= \left| F_{10}^{-1}\big(F_{11}(\theta_1)\big) - F_{10}^{-1}\big(F_{10}(\theta_1)\big) \right| \\
&\leq M_{10} \big| F_{11}(\theta_1) - F_{10}(\theta_1) \big| \\
&\leq M_{10} u_1.
\end{aligned}
$$

47 $\qquad\qquad\qquad\qquad\qquad\qquad\qquad\qquad\qquad\qquad\qquad\qquad\qquad\qquad\qquad\qquad$ □

48 Theorem 1 provides an upper bound on the difference in the false positive rate between the two
49 groups, for any given pair of $(\theta_0, \theta_1)$ such that the false negative rates are the same for the two
50 groups (i.e., satisfies the perfect EOp condition). As discussed in Section 1.2, this upper bound also
51 characterize the additional accuracy loss due to that we have group-dependent thresholds compared
52 to the case with only one threshold for both groups.

### 1.4 Under perfect PE condition

54 For predictive equality (PE) condition, we prove a similar result. That is, assuming we achieve perfect
55 PE condition with

$$\text{FP}_1(\theta_1) = \text{FP}_0(\theta_0), \quad \text{or equivalently} \quad \text{TN}_1(\theta_1) = \text{TN}_0(\theta_0). \tag{9}$$

56 This means that $\theta_0$ and $\theta_1$ satisfies the following condition

$$F_{01}(\theta_1) = F_{00}(\theta_0). \tag{10}$$

57 Equivalently, we have

$$\theta_0 = F_{00}^{-1}\big(F_{01}(\theta_1)\big). \tag{11}$$

58 Under any given pair of $(\theta_0, \theta_1)$ that satisfies (11), the performance error $\mathcal{L}_{per}(\boldsymbol{\theta})$ can be written as

$$
\begin{aligned}
\mathcal{L}_{per}(\boldsymbol{\theta}) &= \left( \frac{n_{01}}{N}\text{FP}_1(\theta_1) + \frac{n_{11}}{N}\text{FN}_1(\theta_1) + \frac{n_{00}}{N}\text{FP}_0(\theta_0) + \frac{n_{10}}{N}\text{FN}_0(\theta_0) \right)^2 \\
&= \left( r_1\text{FN}(\theta_1) + r_0\text{FP}(\theta_1) + \frac{n_{10}}{N}\epsilon_2 \right)^2,
\end{aligned}
$$

59 where

$$\epsilon_2 = \big(\text{FN}_0(\theta_0) - \text{FN}_1(\theta_1)\big).$$

60 Similar to Theorem 1, we can provide an upper bound on $\epsilon_2$ under Assumption 1.

61 **Theorem 2.** *For any given classifier that satisfies Assumption 1 and any given pair of thresholds*
62 *$(\theta_0, \theta_1)$ that satisfies the perfect PE condition, the gap between false-negative rates of the two group*
63 *is upper bounded by*

$$|\epsilon_2| = \big| FN_0(\theta_0) - FN_1(\theta_1) \big| \leq u_1 + C_0 u_0, \tag{12}$$

64 *where $C_0 = \hat{f}_{11} M_{00}$.*

65 *Proof.* The proof is similar to that of Theorem 1. We provide the main steps and omit details that
66 repeat with the proof of Theorem 1. We have

$$
\begin{aligned}
\big| \text{FN}_0(\theta_0) - \text{FN}_1(\theta_1) \big| &= \big| F_{11}(\theta_1) - F_{10}(\theta_0) \big| \\
&\leq \big| F_{11}(\theta_1) - F_{11}(\theta_0) \big| + \big| F_{11}(\theta_0) - F_{10}(\theta_0) \big| \\
&\leq \hat{f}_{11} |\theta_1 - \theta_0| + u_1 \\
&\leq \hat{f}_{11} M_{00} u_0 + u_1.
\end{aligned}
$$

67 $\qquad\qquad\qquad\qquad\qquad\qquad\qquad\qquad\qquad\qquad\qquad\qquad\qquad\qquad\qquad\qquad$ □

Theorem 2 provides an upper bound on the difference in the false negative rate between the two groups, for any given pair of $(\theta_0, \theta_1)$ such that the false positive rates are the same for the two groups (i.e., satisfies the perfect PE condition).

To sum up, Theorem 1 and 2 characterize the upper bound of false positive/negative rate gap between two groups when the false negative/positive rate gap is 0. At the same time, it captures the upper bound of additional accuracy loss due to the two different thresholds for different groups under a perfect fairness (EOp or EP) condition.

## 2 Characterizing the Tradeoff between Accuracy and Fairness

In this section, we prove a theorem to characterize the tradeoff between accuracy and fairness. That is, we start from the perfect EOp or PE conditions and perturb the solution by a small amount. We then bound the difference in the accuracy loss by comparing the perturbed solution with the original solution that satisfies the perfect fairness conditions.

### 2.1 Perturbed EOp condition

To start with, let us consider solutions $(\theta_0, \theta_1)$ that satisfy the perfect EOp condition (5). Under this condition, the optimization problem becomes one dimensional, that is,

$$\theta_1^* = \underset{\theta_1}{\operatorname{argmin}} \, \mathcal{L}_{per}(\theta_1),$$

where

$$\mathcal{L}_{per}(\theta_1) \quad = \quad \left( r_1 \mathrm{FN}_1(\theta_1) + r_0 \mathrm{FP}_1(\theta_1) + \frac{n_{00}}{N} \epsilon_1(\theta_1) \right)^2 \tag{13}$$

and

$$\epsilon_1(\theta_1) = \mathrm{FP}_0(\theta_0) - \mathrm{FP}_1(\theta_1) = F_{01}(\theta_1) - F_{00}\left( F_{10}^{-1}(F_{11}(\theta_1)) \right).$$

From $\theta_1^*$, we can get the corresponding $\theta_0^* = F_{10}^{-1}(F_{11}(\theta_1^*))$. We further denote this optimal accuracy loss value as

$$L^* = \mathcal{L}_{per}(\theta_1^*).$$

Now with the optimal solution $(\theta_0^*, \theta_1^*)$, we investigate the changes in $\mathcal{L}_{per}(\theta_1^*)$ when we perturb the perfect EOp condition and allow a small difference. That is, now consider solution $(\tilde{\theta}_0, \tilde{\theta}_1)$ such that

$$|\mathrm{FN}_1(\theta_1^*) - \mathrm{FN}_1(\tilde{\theta}_1)| \leq \gamma/2, \quad |\mathrm{FN}_0(\theta_0^*) - \mathrm{FN}_0(\tilde{\theta}_0)| \leq \gamma/2. \tag{14}$$

Consequently, the solution $(\tilde{\theta}_0, \tilde{\theta}_1)$ satisfy the following perturbed EOp condition:

$$\left| \mathrm{TP}_1(\tilde{\theta}_1) - \mathrm{TP}_0(\tilde{\theta}_0) \right| = \left| \mathrm{FN}_1(\tilde{\theta}_1) - \mathrm{FN}_0(\tilde{\theta}_0) \right| \leq \gamma. \tag{15}$$

Without loss of generality, we assume that (i) the true positive rate of group 1 is higher than that of group 0, and (ii) the above inequality is binding (because if not binding, then we can always choose a smaller $\gamma$ to make it binding). Thus, we have $\mathrm{TP}_1(\tilde{\theta}_1) - \mathrm{TP}_0(\tilde{\theta}_0) = \gamma$, or equivalently, $\mathrm{FN}_0(\tilde{\theta}_0) - \mathrm{FN}_1(\tilde{\theta}_1) = \gamma$. This gives us

$$\tilde{\theta}_0 = F_{10}^{-1}\left( F_{11}(\tilde{\theta}_1) + \gamma \right). \tag{16}$$

Next, we analyze $\mathcal{L}_{per}(\tilde{\theta}_1)$ by substituting $(\tilde{\theta}_0, \tilde{\theta}_1)$ in (6), which gives us

$$\mathcal{L}_{per}(\tilde{\theta}_1) \quad = \quad \left( r_1 \mathrm{FN}_1(\tilde{\theta}_1) + r_0 \mathrm{FP}_1(\tilde{\theta}_1) + \frac{n_{10}}{N}\gamma + \frac{n_{00}}{N}\tilde{\epsilon}_1(\tilde{\theta}_1) \right)^2, \tag{17}$$

where

$$\tilde{\epsilon}_1(\tilde{\theta}_1) = \mathrm{FP}_0(\tilde{\theta}_0) - \mathrm{FP}_1(\tilde{\theta}_1) = F_{01}(\tilde{\theta}_1) - F_{00}\left( F_{10}^{-1}\left( F_{11}(\tilde{\theta}_1) + \gamma \right) \right).$$

We denote the optimal value for this perturbed version as $\tilde{\theta}_1^*$, and its corresponding loss value as

$$\tilde{L}^* = \mathcal{L}_{per}(\tilde{\theta}_1^*).$$

Furthermore, from (14), we have

$$|\mathrm{FN}_1(\theta_1^*) - \mathrm{FN}_1(\tilde{\theta}_1^*)| = |F_{11}(\theta_1^*) - F_{11}(\tilde{\theta}_1^*)| \le \gamma/2. \tag{18}$$

Under Assumption 1, we have

$$
\begin{aligned}
|\theta_1^* - \tilde{\theta}_1^*| &= \left|F_{11}^{-1}(F_{11}(\theta_1^*)) - F_{11}^{-1}(F_{11}(\tilde{\theta}_1^*))\right| \\
&\le M_{11}\left|F_{11}(\theta_1^*) - F_{11}(\tilde{\theta}_1^*)\right| \\
&= M_{11}\gamma/2.
\end{aligned}
$$

## 2.2 Theorem 3 and its proof

We are ready to compare $\mathcal{L}_{per}(\theta_1^*)$ and $\mathcal{L}_{per}(\tilde{\theta}_1^*)$. The latter loss should be no larger than the former since we relaxed the perfect EOp condition (constraint) in the optimization, i.e., $L^* \ge \tilde{L}^*$.

**Theorem 3.** *Under Assumption 1 and condition* (14),

$$\mathcal{L}_{per}(\theta_1^*) - \mathcal{L}_{per}(\tilde{\theta}_1^*) \le C\gamma,$$

*where* $C = 2L^*\left(\frac{r_1}{2} + r_0\frac{\hat{f}_{01}M_{11}}{2} + \frac{n_{00}}{N}\left(\hat{f}_{00}M_{10} + \frac{\hat{\epsilon}_1'M_{11}}{2}\right) + \frac{n_{10}}{N}\right)$, *and* $\hat{\epsilon}_1' = \max \tilde{\epsilon}_1'$ *is the maximum of the derivative of* $\tilde{\epsilon}_1$.

*Proof.* We have that

$$
\begin{aligned}
&\mathcal{L}_{per}(\theta_1^*) - \mathcal{L}_{per}(\tilde{\theta}_1^*) \\
\le\ & 2L^*\bigg|r_1\mathrm{FN}_1(\theta_1^*) + r_0\mathrm{FP}_1(\theta_1^*) + \frac{n_{00}}{N}\epsilon_1(\theta_1^*) \\
&\quad - \left(r_1\mathrm{FN}_1(\tilde{\theta}_1^*) + r_0\mathrm{FP}_1(\tilde{\theta}_1^*) + \frac{n_{10}}{N}\gamma + \frac{n_{00}}{N}\tilde{\epsilon}_1(\tilde{\theta}_1^*)\right)\bigg| \\
\le\ & 2L^*\left(r_1\gamma/2 + r_0|\mathrm{FP}_1(\theta_1^*) - \mathrm{FP}_1(\tilde{\theta}_1^*)| + \frac{n_{00}}{N}|\epsilon_1(\theta_1^*) - \tilde{\epsilon}_1(\tilde{\theta}_1^*)| + \frac{n_{10}}{N}\gamma\right),
\end{aligned}
$$

where we further have that

$$
\begin{aligned}
|\mathrm{FP}_1(\theta_1^*) - \mathrm{FP}_1(\tilde{\theta}_1^*)| &= |F_{01}(\theta_1^*) - F_{01}(\tilde{\theta}_1^*)| \\
&\le \hat{f}_{01}|\theta_1^* - \tilde{\theta}_1^*| \\
&\le \hat{f}_{01}M_{11}\gamma/2,
\end{aligned}
$$

and

$$
\begin{aligned}
\left|\epsilon_1(\theta_1^*) - \tilde{\epsilon}_1(\tilde{\theta}_1^*)\right| &\le \left|\epsilon_1(\theta_1^*) - \tilde{\epsilon}_1(\theta_1^*)\right| + \left|\tilde{\epsilon}_1(\theta_1^*) - \tilde{\epsilon}_1(\tilde{\theta}_1^*)\right| \\
&\le \left|F_{00}(F_{10}^{-1}(F_{11}(\theta_1^*))) - F_{00}(F_{10}^{-1}(F_{11}(\theta_1^*) + \gamma))\right| + \hat{\epsilon}_1'M_{11}\gamma/2 \\
&= (\hat{f}_{00}M_{10} + \hat{\epsilon}_1'M_{11}/2)\gamma.
\end{aligned}
$$

Here, $\hat{\epsilon}_1' = \max \tilde{\epsilon}_1'$ is the maximum of the derivative of $\tilde{\epsilon}_1$. Combining all the terms in front of $\gamma$ gives us the desired upper bound. $\square$

Theorem 3 quantifies the decrease in accuracy loss (i.e., the improvement in accuracy) when we allow a gap of true positive rates between two groups (i.e., relaxation from the perfect EOp condition).

Repeating the analysis for the perturbed PE condition, we can obtain a similar bound for the changes in the accuracy loss function. We omit the details here in the interest of space.

# 3 Experimental Details

## 3.1 Comparing Methods

We compared our method with multiple state-of-the-art methods to verify our work. The details about the comparing methods are as below:

- **Learning fair representations for kernel models** (abbreviated as FGP) [11]: a pre-processing method to learn representation focusing on kernel-based models. The fair model that satisfies certain fairness criterion is obtained by Bayesian learning from fair Gaussian process (FGP) prior.

- **Fairness confusion tensor** (abbreviated as FACT) [5]: a post-processing model that minimize the least-squares accuracy-fairness optimality problem based on confusion tensor.

- **Adversarial de-biasing** (abbreviated as AdvDeb) [12]: an in-processing model that mitigates the conflicting gradient directions in utility and fairness objectives by projecting one gradient to another to remove the opposite direction.

- **Calibrated equal odds post-processing** (abbreviated as CEOPost) [9]: a post-processing method that minimizes the disparity in the predicted probability to the preferred class among different sensitive groups, while maintaining the calibration condition in a relaxed condition.

- **Equality of opportunity in supervised learning** (abbreviated as Eq.Odds) [3]: a post-processing method that learns the threshold to yield the equalized odds/opportunity between different demographic by exploring the intersection of achievable true positive rates and false positive rates.

- **Learning adversarially fair and transferable representations** (abbreviated as LAFTR) [8]: a fair representation learning model that adopts fairness metrics as the adversarial objectives and analyze the balance between utility and fairness.

- **Baseline**: For CelebA dataset, we use ResNet50 [4] as a reference because we input second last layer(2048 features) of ResNet to all methods. For other tabular datasets, logistic regression is used as all other methods except for FGP and LAFTR are based on logistic regression.

If the hyperparameter is adjustable in the listed methods, we report the result with the fairness coefficient that has an accuracy closest to the average accuracy in the coefficient sweep to balance utility and fairness.

## 3.2 Evaluation Metrics

In the experiments, we evaluate the methods on four fairness and two performance measures. Four fairness metrics are as below:

- **Equal Opportunity** (abbreviated as EOp) [3] : This measures absolute difference of favorable prediction given positive label.

$$|P(\hat{Y} = 1|Y = 1, A = 1) - P(\hat{Y} = 1|Y = 1, A = 0)|.$$

- **Equalized Odds** (abbreviated as EOd) [3] : This measures the difference between the probability given the true labels.

$$|P(\hat{Y} = 1|Y = 1, A = 1) - P(\hat{Y} = 1|Y = 1, A = 0)|+$$
$$|P(\hat{Y} = 1|Y = 0, A = 1) - P(\hat{Y} = 1|Y = 0, A = 0)|.$$

- **Balanced Accuracy Difference** (abbreviated as BD) : This measures difference between balanced accuracy between the groups.

$$|P(\hat{Y} = 1|Y = 1, A = 1) + P(\hat{Y} = 0|Y = 0, A = 1)|$$
$$- |P(\hat{Y} = 1|Y = 1, A = 0) + P(\hat{Y} = 0|Y = 0, A = 0)|.$$

If BD and EOd has the same value, it indicates that both TPR and TNR are higher in a certain sensitive group. However, if the gap between the two terms is large, we can interpret as the classifier is more biased as a group with higher TPR has lower TNR. This is more unfair as a sample from the privileged group is more likely to be falsely and correctly predicted as positive output. EOp is a partial measure of EOd as it only measures the difference from a favorable class.

| CelebA | | | | |
|---|---|---|---|---|
| Model | GSTAR | FGP | FACT | CEOPost |
| Time | 0.287 | - | 0.067 | 0.077 |
| Model | DIR | Eq.Odds | LAFTR | AdvDeb |
| Time | 183.20 | 0.062 | 107.04(min) | 303.15 |
| Adult | | | | |
| Model | GSTAR | FGP | FACT | CEOPost |
| Time | 0.29 | 51.28 | 0.055 | 25.61 |
| Model | DIR | Eq.Odds | LAFTR | AdvDeb |
| Time | 168 | 0.037 | 53.04(min) | 102.00 |
| Compas | | | | |
| Model | GSTAR | FGP | FACT | CEOPost |
| Time | 0.292 | 43.74 | 0.035 | 8.3 |
| Model | DIR | Eq.Odds | LAFTR | AdvDeb |
| Time | 123.20 | 0.034 | 57.04(min) | 15.45 |
| German | | | | |
| Model | GSTAR | FGP | FACT | CEOPost |
| Time | 0.271 | 7.08 | 0.0257 | 2.64 |
| Model | DIR | Eq.Odds | LAFTR | AdvDeb |
| Time | 1.68 | 0.034 | 56.51(min) | 2.17 |

Table 1: Computational time (in seconds) for all comparing fairness methods for each dataset.

- **Absolute (1 - Disparate Impact)** (abbreviated as 1-DIMP) [1] : This measures ratio of the probability of the favorable prediction given a protected group.

$$\left| 1 - \frac{P(\hat{Y} = 1 | A = 1)}{P(\hat{Y} = 1 | A = 0)} \right|.$$

We evaluate performance of the methods with two metrics.

- **Balanced Accuracy** (abbreviated as BA) : This measures average between true positive rate and true negative rate. Compared to the traditional accuracy, this measure effectively shows the whether the classifier is focusing on the performance of a certain class when the dataset is unbalanced.

$$\frac{1}{2} \left( P(\hat{Y} = 1 | Y = 1) + P(\hat{Y} = 0 | Y = 0) \right).$$

- **Accuracy** (abbreviated as ACC) : This measures traditional classification accuracy of the method.

## 3.3 Dataset Description

We evaluate the methods on four fairness datasets. The goal for all datasets is binary classification on binary sensitive feature. The details of the datasets are as below:

- **CelebA image dataset**[1] [7]: The data consists of 202,599 face images in diverse demographics. The images are annotated with 40 attributes (face shape, skin tone, smiling, etc.). Similar to Quadrianto *et al.* [10], the goal is to predict whether a person in the image is attractive or not. The feature *sex* is used as the sensitive feature.

- **Adult** dataset from the UCI repository [6] contains 48,842 instances described by 14 features (workclass, age, education, sex, race, *etc*) with the goal of the income prediction whether a person's income exceeds 50K USD per year. The feature *sex* is used as the sensitive feature.

- **COMPAS**[2](Correctional Offender Management Profiling for Alternative Sanctions) dataset includes 6,167 samples described by 401 features with the target of recidivism prediction

---

[1]http://mmlab.ie.cuhk.edu.hk/projects/CelebA.html
[2]https://github.com/propublica/compas-analysis

with the label showing if each person gets rearrested within two years. The feature *race* is used as the sensitive feature for this dataset.

- **German** credit dataset from the UCI repository [2] contains 1,000 samples described by 20 features. The goal is to predict the credit risks. The feature *sex* is used as the sensitive feature.

## 3.4 Computational Cost

In Table 1, we describe the computational time for each method on each dataset. By introducing estimated PDF functions for post-processing, we outperform other methods except Eq.Odds [3] and FACT [5]. As they both only utilize the entries of the confusion matrix to find optimal mixing rate in their methods, they have less computation than ours. However, as we discussed in the main paper, we explore better feasible region than theirs by group-specific thresholding that results better in both fairness and performance by sacrificing little efficiency, yet outperforms most of the other works.