# OpenReview forum: "Group-Aware Threshold Adaptation for Fair Classification"
_NeurIPS.cc/2021/Conference — NeurIPS 2021 Submitted_

### Official Review · Reviewer_MyCa · 2021-07-11

**Rating:** 5
**Confidence:** 4

**Summary:**

This paper proposes a threshold modification method to post-process the trained classifiers. The threshold learning is formulated as an optimization task and an efficient technique is proposed to learn the thresholds for groups. The proposed adaptive threshold learning is able to work with multiple fairness constraints.

**Ethical Concerns:**

Yes

**Main Review:**

originality: fair
quality: good
clarity: good
significance: good

Strength:

1. The proposed method learns adaptive thresholds for various groups to achieve fairness. The learning technique is efficient. by using Taylor expansion. The efficient algorithm works for multiple fairness constraints.
2. There theoretical results in Appendix are very interesting.

Weakness:

1. The proposed method requires a pre-defined parametric probability distribution. The requirement is kind of difficult for many real-world applications.  It is unclear whether the proposed method is sensitive to the selection of distribution.
2. The alternative optimization method is fast but there is a lack of guarantee for the optimum. It is unclear how close the approximation to the optimal solution.

Overall, the contribution of the proposed idea (threshold adjusting) is interesting. The attempt to develop an efficient learning algorithm is also interesting.  My major concern is the efficient algorithm incurs approximation errors due to the Taylor expansion. The learning algorithm requires the density function. It is unclear how the proposed alternative optimization is sensitive to the density estimation.

## Post response

I truly appreciate the response. It is informative and helpful for better understanding the manuscript.

I had two major concerns on the density assumption and convergence. The convergence analysis is more like a practical solution to an optimization task. It shows how to approximate a solution but raises some new concerns such as how sensitive the solution is w.r.t. the relaxation and Taylor expansion. The empirical results show the sensitivity of accuracy and fairness w.r.t. the density estimation. I understand the theoretical challenge behind but I think it deserves further investigation as the optimization later requires a good estimation. In summary, my opinions and rating are unchanged.

**Time Spent Reviewing:**

3

---

> ### Author Response · Authors · 2021-08-10
> **Author respond to answer the questions by the reviewer.**
>
> We sincerely thank the reviewer for taking the time to review our paper and provide constructive comments.
>
> **[Quality of estimated distribution]**
>
> It is true that the GSTAR performance relies on the estimated distribution. We empirically found that the distribution of logits resembles some parametric distributions. Thus, we estimate the distribution by choosing from generally used parametric distributions such as Gaussian, gamma, etc., by measuring log-likelihood in the training data. However, this can be extended to a wide range of other distributions, even non-parametric distributions.
>
> For further analysis, we sweep the parameters of parametric distribution to see the sensitivity of GSTAR to the estimation quality. Take COMPAS dataset as an example, the best estimate (i.e., smallest NLL) of group $(y = 0, a = 0)$ with Student's t-distribution has parameters of df = 2.235, loc = -0.567, scale = 0.756 based on scipy package. Then to generate distributions with varying estimation qualities, we add $\alpha \in [-0.1, 100]$ to the scale of this distribution. Then we can get the variation of the parametric distribution as in the figure [link-1]. Given the variations, we observed the trend of fairness and performance as in figure [link-2].
>
> From figures in link-1 and link-2, we observe the change of NLL (black solid), fairness violation (lower value is better), and accuracy (higher value is better) with varying noise ($\alpha$, x-axis). The color of lines follow the main paper. Dashed lines indicate the quantity of baseline model ($\theta = 0$). In the figures, we observed that the accuracy is the most sensitive to the change of negative log-likelihood, while fairness is relatively stable w.r.t. the estimation quality. We will add the corresponding figures in the final paper. However, we assume the estimation is reliable and the guarantee on the estimation reliability is beyond our focus of this paper.
>
> **[Convergence analysis]**
>
> Our objective function and the optimization solution algorithm belong to the family of Gauss-Newton algorithm to solve Nonlinear Least Squares Problem (NLSP). To specify, NLSP is to solve
> $$\min_{\mathbf{\theta}} ||r(\mathbf{\theta})||^2_2,$$
> where the decision variables, $\mathbf{\theta}$, is an $n$-dimensional real vector and the objective function $r$ is an $m$-dimensional real vector function of $\mathbf{\theta}$. Connecting to our setting and using two groups as an example, our decision variables is the two-dimensional vector $\mathbf{\theta}=(\theta_0, \theta_1)$ for group 0 and group 1, and our objective function is the following 2-dimensional real vector function:
> $$r(\mathbf{\theta}) = \big( r_1(\mathbf{\theta}), r_2(\mathbf{\theta}) \big)$$
> with $$ r_1(\mathbf{\theta}) = r_1(\theta_0, \theta_1) = \frac{n_{01}}{N}{FP_1}(\theta_1) + \frac{n_{11}}{N}{FN_1}(\theta_1) + \frac{n_{00}}{N}{FP_0}(\theta_0) + \frac{n_{10}}{N}{FN_0}(\theta_0), $$
> $$ r_2(\mathbf{\theta}) = r_2(\theta_0, \theta_1) = \sqrt{\lambda}\left(TP_1(\theta_1) - TP_0(\theta_0)\right) $$
> when taking the EOp constraint.
> The $L_2$ norm $||r(\mathbf{\theta})||^2_2 = r_1(\mathbf{\theta})^2 + r_2(\mathbf{\theta})^2$ recovers the objective function in Equation (2).
>
> A classic family of algorithms to solve NLSP is the Gauss-Newton Method. The main idea is to convert the nonlinear optimization problem to a linear least square problem using Taylor expansion. That is, the parameter values are calculated in an iterative fashion with
> $$\theta_{j} \approx \theta_{j}^{k+1} = \theta_{j}^{k} +\Delta_j,$$
> in the $k$-th iteration number, with the vector of increments $\Delta=\lbrace\Delta_j\rbrace = \lbrace \theta_{j}^{k+1} - \theta_{j}^{k} \rbrace$ (also known as the shift vector). We linearize each component in the $f$ function to a first-order Taylor polynomial expansion as
> $$r_i (\mathbf{\theta}) \approx r_i(\mathbf{\theta}^k) + \sum_{j} \frac{\partial r_i(\mathbf{\theta})}{\partial\theta_j} \Delta_j$$
> with $\mathbf{\theta}^k =(\theta_0^{k}, \theta_1^k)$. Plugging this linearized equation into the objective function, we get the usual least square problem. Then, the optimal solution can be obtained as
> $$\Delta = - (J^T J)^{-1} J^T f(\mathbf{\theta}^k),$$
> where $J=\{J_{ij}\}$ with $J_{ij} = \{\frac{\partial r_i(\mathbf{\theta})}{\partial\theta_j}\}$ is the Jacobian. Note that in the GSTAR algorithm, we ignore the terms for $j\neq i$ in the Taylor expansion. Thus, in calculating $J^T J$, we only kept the diagonal terms
> $$\left(\frac{\partial  r_1(\mathbf{\theta})}{\partial\theta_j}\right)^2 + \left(\frac{\partial  r_2(\mathbf{\theta})}{\partial\theta_j}\right)^2$$
> for $j=0,1$. Plugging in the form of $r_1$ and $r_2$ as specified above, we achieve the solution provided in Equation (10) in the main paper.
>
> There is a long history of studying the approximations to the nonlinear problems in Gauss–Newton algorithm and convergence property, e.g., see [1]. The convergence of the algorithm is generally not guaranteed, e.g., if the initial solution is far from the true optimal or $J^T J$ is ill-conditioned. In other words, the convergence of the algorithm heavily depends on the density estimation $f(\cdot)$. We state the following sufficient conditions from [1] to guarantee the convergence of the algorithm. The following assumptions are made in order to establish the theory.
>
> * A1. There exists $\theta^*$ such that $J^T(\theta^*)r(\theta^*) = 0$;
>
> * A2. The Jacobian at $\theta^*$ has full rank.
>
> We state Theorem. 4 from [1] on the sufficient conditions for convergence.
>
> **Theorem 4 [1]** _Assume that the estimated density function $f(\cdot)$ satisfy assumptions A1 and A2. Further, $f(\cdot)$ satisfies that_ $$||Q(\theta^k)(J^T J)^{-1}(\theta^k)||_2 \leq \eta $$
> _for some constant $\eta\in[0,1)$ for each iteration $k$, where $Q(\theta)$ denotes the second order terms $\sum_i r_i(\theta) \nabla^2 r_i(\theta)$. Then as long as the initial solution is sufficiently close to the true optimal with_ $||\theta^0 -\theta^*||_2 \leq \epsilon$, _the sequence of Gauss-Newton iterates $\{\theta^k\}$ converges to $\theta^*$._
>
> It is known that for general function $f(\cdot)$ such as estimates from a neural network, the above sufficient conditions that guarantee convergence do not necessarily hold. As a result, protection against divergence is essential. In our numerical experiments, we adopt a commonly used, simple protection, the shift-cutting method. That is, we to reduce the length of the shift vector $\Delta$ by a fraction $\eta$. In other words, the update becomes $$ \theta_j^{k+1} = \theta_j^k + \eta \Delta_j. $$
>
> Thank you again for your insightful comments and we will include all the materials delivered in this response in the final draft.
>
> **[Reference]**
>
> [1] Gratton, Serge, Amos S. Lawless, and Nancy K. Nichols. ``Approximate Gauss–Newton methods for nonlinear least squares problems.'' SIAM Journal on Optimization 18.1 (2007): 106-132.
>
> [link-1] https://drive.google.com/file/d/1IysP_Ye5MCwcmcF2odaZuWrf_Pk6SKgw/view?usp=sharing
>
> [link-2] https://drive.google.com/file/d/1WNRas4rLrNBqaBnU5f2yC334EKmu30_u/view?usp=sharing

---

### Official Review · Reviewer_qnqF · 2021-07-14

**Rating:** 7
**Confidence:** 4

**Summary:**

The authors propose a new post-processing group-aware threshold search procedure for fair classification. The procedure is efficient, amenable to different types of underlying classification models, and is empirically shown to be working with multiple fairness criteria. The authors also derive  Pareto frontier of their model for the fairness-accuracy trade-off. Lastly, the authors conclude with experiments on four datasets showing the efficacy of their method.

**Limitations And Societal Impact:**

Yes.

**Main Review:**

Strengths:
1. Relevant problem to the NeurIPS community.
2. Rigorous treatment to an important problem.
3. Experimental evaluation is good that includes multiple datasets along with relevant baselines.

Weaknesses:
1. Novelty: The idea of using multiple thresholds for different demographic groups seems to be an obvious thing to do. Prior work may not have done analysis of the type that is provided in this paper, but my guess is it could have been used in the previous papers (maybe as baselines). I would let other reviewers who are more aware of the fairness literature to decide on the novelty of the paper.
2. The authors have not provided any convergence analysis of the proposed procedure. Is it guaranteed to converge to a good solution (good thresholds)?
2. Minor point: The entire theoretical analysis has been pushed to appendix including the results on Pareto frontier. If the paper is accepted, I would encourage authors to bring some results in the main paper.

Detailed Comments:
Overall, I liked the paper and do not have any major concern other than novelty of the idea. The following points may further help to clarify/improve the paper.

1. In related works, especially, lines 74-79, can you please specify how your work is different from prior work?

2. Why is there a square in equation 3? Is it because it helps in derivation? Can we also use weighted measures for L_per and L_fair as done in the following papers:
Hiranandani, Gaurush, et al. "Fair Performance Metric Elicitation." NeurIPS (2020). -- Definition 1, Equation (6)
Hiranandani, Gaurush, et al. "Quadratic Metric Elicitation with Application to Fairness." arXiv preprint arXiv:2011.01516 (2020). -- Definition 4, equation (13)

3. Equation (8) does not contain any subscripts; whereas, the terms below line 180 do. Can you please clarify?

4. As mentioned before, the authors have not provided any convergence analysis of the proposed procedure. Is it guaranteed to converge to a good solution (good thresholds)?
Also, what happens to their procedure in case of multiple sensitive groups? Does a coordinate wise optimization for thresholds work in this case?

5. The graphs are not readable. Authors can increase the legend size, if possible, remove some trivial baselines to make the graphs more readable.

6. There are good insights in lines 272-275.

7. As the authors mention in their limitations that their method may not work if the underlying models gives over-confident probability distributions. One suggestion to the authors is that they can use calibration or label-smoothening in that case.

Minor:
The authors can remove bold  text while explaining baselines.

**Time Spent Reviewing:**

3

---

> ### Author Response · Authors · 2021-08-10
> **Author respond to answer the questions by the reviewer.**
>
> We sincerely thank the reviewer for taking the time to review our paper and provide constructive comments.
>
> **[Novelty and related works]**
>
> The work of [Corbett-Davies et al., 2017] differs from ours as their thresholds are not trainable variables, which are instead determined based on the statistics of the training data. In contrast, our GSTAR model learns the adaptive thresholds via optimization which can adapt to various fairness notions and find the optimal thresholds to balance between accuracy and fairness.
>
> [Hardt et al., 2016] propose a post-processing method given the output of a vanilla classifier. Let $\hat Y$, and $\tilde Y$ as an output of vanilla classifier and their proposed method respectively. They employ randomization such that they flip the predicted labels with the probability $P(\tilde Y = 1|\hat Y = 1 , A = a)$ and $P(\tilde Y = 1|\hat Y = 0 , A = a)$, which is called mixing rate to achieve equalized odds/equal opportunity.
> The random flipping flips the output of randomly selected samples to equalize the true positive rate between sensitive groups, which may cause diverse societal problems (e.g., lawsuit) as some {samples with high confidence (i.e., easy samples)} could be flipped.
> Also, GSTAR is not confined to Equalized Odds or Equal opportunity and generalizes to diverse fairness notions, and can incorporate multiple constraints.
>
> Similarly, [Kamiran et al., 2012] propose to make a prediction based on the confidence of a vanilla classifier. For a given threshold $\theta$, if a sample x is in a critical region $\max(P(\hat Y = 1|X = x, A = a), P(\hat Y = 1|X = x, A = a)) < \theta$,
> then adjust $\tilde Y$ to a favorable/unfavorable label for unprivileged/privileged group. This method considers confidence so that it is free from the problem that [Hardt et al., 2016] had, but they didn’t propose how to obtain the threshold $\theta$.
>
> The main difference between GSTAR (ours) and FACT [Kim et al., 2020] is that GSTAR leverages more information about the black-box classifier than FACT. To construct the confusion tensor, FACT uses the discrete output (i.e., the predicted label) from the black-box classifier to compare with the true label, and obtain the threshold using the method proposed in [Hardt et al., 2016]. However, GSTAR uses the distribution of the logits (with continuous values) which contains more information. Fig. 2 of the main paper validates that GSTAR achieves a better accuracy-fairness trade-off than FACT.
>
> **[Squared term in objective, and comparison with Hiranandani et al., 2020]**
>
> [Hiranandani et al., NeurIPS 2020] use the L1 norm. [Hiranandani et al., arXiv 2020] use a weighted average between a L1 norm on misclassification and a L2 norm on the fairness metrics. We use L2 norm for both classification and fairness to form a nonlinear least square problem.
>
> In the method of [Hiranandani et al., 2020], they weight the violation and misclassification rate group-wise and use L1 norm.
> Ours can be considered as a unit weight for each group and calculate the best thresholds efficiently.
> We can incorporate such weight vectors to GSTAR with relative preference feedback by the needs of the user (oracle).
>
> Thank you for the suggestion. We will leave the incorporation of weight vectors in GSTAR to future work.
>
> **[Convergence analysis]**
>
> Our objective function and the optimization solution algorithm belong to the family of Gauss-Newton algorithm to solve Nonlinear Least Squares Problem (NLSP). To specify, NLSP is to solve
> $$\min_{\mathbf{\theta}} ||r(\mathbf{\theta})||^2_2,$$
> where the decision variables, $\mathbf{\theta}$, is an $n$-dimensional real vector and the objective function $r$ is an $m$-dimensional real vector function of $\mathbf{\theta}$. Connecting to our setting and using two groups as an example, our decision variables is the two-dimensional vector $\mathbf{\theta}=(\theta_0, \theta_1)$ for group 0 and group 1, and our objective function is the following 2-dimensional real vector function:
> $$r(\mathbf{\theta}) = \big( r_1(\mathbf{\theta}), r_2(\mathbf{\theta}) \big)$$
> with $$ r_1(\mathbf{\theta}) = r_1(\theta_0, \theta_1) = \frac{n_{01}}{N}{FP_1}(\theta_1) + \frac{n_{11}}{N}{FN_1}(\theta_1) + \frac{n_{00}}{N}{FP_0}(\theta_0) + \frac{n_{10}}{N}{FN_0}(\theta_0), $$
> $$ r_2(\mathbf{\theta}) = r_2(\theta_0, \theta_1) = \sqrt{\lambda}\left(TP_1(\theta_1) - TP_0(\theta_0)\right) $$
> when taking the EOp constraint.
> The $L_2$ norm $||r(\mathbf{\theta})||^2_2 = r_1(\mathbf{\theta})^2 + r_2(\mathbf{\theta})^2$ recovers the objective function in Equation (2).
>
> A classic family of algorithms to solve NLSP is the Gauss-Newton Method. The main idea is to convert the nonlinear optimization problem to a linear least square problem using Taylor expansion. That is, the parameter values are calculated in an iterative fashion with
> $$\theta_{j} \approx \theta_{j}^{k+1} = \theta_{j}^{k} +\Delta_j,$$
> in the $k$-th iteration number, with the vector of increments $\Delta=\lbrace\Delta_j\rbrace = \lbrace \theta_{j}^{k+1} - \theta_{j}^{k} \rbrace$ (also known as the shift vector). We linearize each component in the $f$ function to a first-order Taylor polynomial expansion as
> $$r_i (\mathbf{\theta}) \approx r_i(\mathbf{\theta}^k) + \sum_{j} \frac{\partial r_i(\mathbf{\theta})}{\partial\theta_j} \Delta_j$$
> with $\mathbf{\theta}^k =(\theta_0^{k}, \theta_1^k)$. Plugging this linearized equation into the objective function, we get the usual least square problem. Then, the optimal solution can be obtained as
> $$\Delta = - (J^T J)^{-1} J^T f(\mathbf{\theta}^k),$$
> where $J=\{J_{ij}\}$ with $J_{ij} = \{\frac{\partial r_i(\mathbf{\theta})}{\partial\theta_j}\}$ is the Jacobian. Note that in the GSTAR algorithm, we ignore the terms for $j\neq i$ in the Taylor expansion. Thus, in calculating $J^T J$, we only kept the diagonal terms
> $$\left(\frac{\partial  r_1(\mathbf{\theta})}{\partial\theta_j}\right)^2 + \left(\frac{\partial  r_2(\mathbf{\theta})}{\partial\theta_j}\right)^2$$
> for $j=0,1$. Plugging in the form of $r_1$ and $r_2$ as specified above, we achieve the solution provided in Equation (10) in the main paper.
>
> There is a long history of studying the approximations to the nonlinear problems in Gauss–Newton algorithm and convergence property, e.g., see [1]. The convergence of the algorithm is generally not guaranteed, e.g., if the initial solution is far from the true optimal or $J^T J$ is ill-conditioned. In other words, the convergence of the algorithm heavily depends on the density estimation $f(\cdot)$. We state the following sufficient conditions from [1] to guarantee the convergence of the algorithm. The following assumptions are made in order to establish the theory.
>
> * A1. There exists $\theta^*$ such that $J^T(\theta^*)r(\theta^*) = 0$;
>
> * A2. The Jacobian at $\theta^*$ has full rank.
>
> We state Theorem. 4 from [1] on the sufficient conditions for convergence.
>
> **Theorem 4 [1]** _Assume that the estimated density function $f(\cdot)$ satisfy assumptions A1 and A2. Further, $f(\cdot)$ satisfies that_ $$||Q(\theta^k)(J^T J)^{-1}(\theta^k)||_2 \leq \eta $$
> _for some constant $\eta\in[0,1)$ for each iteration $k$, where $Q(\theta)$ denotes the second order terms $\sum_i r_i(\theta) \nabla^2 r_i(\theta)$. Then as long as the initial solution is sufficiently close to the true optimal with_ $||\theta^0 -\theta^*||_2 \leq \epsilon$, _the sequence of Gauss-Newton iterates $\{\theta^k\}$ converges to $\theta^*$._
>
> It is known that for general function $f(\cdot)$ such as estimates from a neural network, the above sufficient conditions that guarantee convergence do not necessarily hold. As a result, protection against divergence is essential. In our numerical experiments, we adopt a commonly used, simple protection, the shift-cutting method. That is, we to reduce the length of the shift vector $\Delta$ by a fraction $\eta$. In other words, the update becomes $$ \theta_j^{k+1} = \theta_j^k + \eta \Delta_j. $$
>
> **[Extension to multiple sensitive group scenario]**
>
> Our algorithm can be extended to dealing with multiple sensitive groups naturally. In this case, the decision variables just becomes a multi-dimensional vector $(\theta_1, \theta_2, \dots, \theta_n)$. The objective function also incorporates additional terms that capture the corresponding fairness metrics for multiple groups. Then, our Gauss-Newton type method still applies with a multi-variable Taylor expansion, and the solution will follow the same form as delivered in convergence analysis above.
>
> **[Typo]**
>
> Thank you for pointing out the typo in equation (8). There should be subscript “1” as other terms in the equation.
>
> We will improve the readability of the figures and include all the materials delivered in this response in the final draft.
>
> **[Reference]**
>
> [1] Gratton, Serge, Amos S. Lawless, and Nancy K. Nichols. "Approximate Gauss–Newton methods for nonlinear least squares problems.'' SIAM Journal on Optimization 18.1 (2007): 106-132.

---

> > ### Comment · Reviewer_qnqF · 2021-08-25
> > **Re: Post Rebuttal**
> >
> > I thank the authors for providing clarifications on my questions. Also, it is good to see the convergence analysis. I would request authors to add this analysis if the paper is accepted. I stand by my rating and am inclined to accept this paper.

---

> > > ### Author Response · Authors · 2021-08-30
> > > **Author response for post rebuttal**
> > >
> > > We thank the reviewer for recognizing our work and providing valuable suggestions for the revision of our paper. We really appreciate the reviewer's time and effort in reviewing our paper. We will add the convergence analysis in our final paper.

---

### Official Review · Reviewer_5UxG · 2021-07-15

**Rating:** 5
**Confidence:** 4

**Summary:**

This paper introduces a method for finding fair and optimal thresholds for binary classification. It considers the mean squared error as the objective function. Instead of solving a constrained optimization problem, they add a  fairness loss to the objective function.

Then using Taylor approximation,  the optimization problem is reduced to an optimization function with a quadratic utility. The proposed method has low complexity ($\mathcal{O}(n+T)$). Moreover, it improves the fairness accuracy tradeoff as compared to existing works.

**Limitations And Societal Impact:**

Yes, They have.

**Main Review:**

While Algorithm 1 is pretty clear, the experiment results are not clear at all.

For instance, in Figure 2, LAFTR is an adversarial method for fair learning. LAFTR works for different fairness notions (e.g., equal opportunity, statistical party (SP), etc.). However, in figure 2, is not clear which fairness notions have been considered for LAFTR.  Also, there are different points for a single algorithm in each plot. What does that mean?
The x-axis in figure 2 is EOd. What does this mean? EOd imposes two constraints. how do you measure it in one dimension?

The same issue exists in Figure 3.  Eq.Odds [7] can be used to satisfy EOd or EOp.  LAFTR can be used for EOD EOp and SP. Which constraint are you using when training LAFTR or Eq.Odds. If LAFTR is trained for satisfying EOp, and GSTAR is designed to satisfy EOp, then we cannot compare the fairness accuracy tradeoff.

You should clarify in detail that what each element in figure 2 and 3 mean. Without clarification, I am not able to compare your method with other methods.  It is very likely that I change my score after the authors' response.

**Time Spent Reviewing:**

5 hours

---

> ### Author Response · Authors · 2021-08-10
> **Author respond to answer the questions by the reviewer.**
>
> We sincerely thank the reviewer for taking the time to review our paper and provide constructive comments.
>
> **[Experimental setup]**
>
> For experimental setup, all comparing methods apply EOd as the fair constraint, thus we compare them via EOd in Fig. 2. Both the Pareto frontier from GSTAR and FACT are derived based on EOd constraint for a fair comparison. To report FACT results, we follow the setup in Section G.3 of the FACT paper [Kim et al., 2020], which does not require $\lambda$. For our setup, we estimate $f_{ya}$ and optimize $\theta_a$ from the training data, and report evaluation results (with the $\theta_a$ learned from training data) on the testing data. We use the same $\lambda$ to incorporate multiple fairness constraints for simplicity, but different $\lambda$ can be introduced individually.
>
> Fig. 2 illustrates Pareto frontiers by sampling 50 $\lambda$ values by sweeping in $[10^{-2},10^7]$ with equal logspace. Similarly for comparing methods, we sweep hyperparameters (e.g, weights for each term in the objective function) to visualize Pareto frontiers. Fig. 3 takes $\lambda$ or hyperparameter values from the upper-right point of the Pareto frontiers in Fig. 2, which indicates the best fairness-accuracy tradeoff for each method. Fig. 3 presents the 5 runs with the setup chosen based on the Pareto frontier to show the consistency of the performance of each model.
>
> Thank you again for your insightful comments and we will include all the materials delivered in this response in the final draft.

---

> > ### Comment · Reviewer_5UxG · 2021-08-17
> > **More questions for the authors**
> >
> > I still believe that experiments are not clear. A few more questions about figure 2:
> >
> > 1. What does mathematically mean if EOd is equal to 0.1?
> >
> > 2. Eq.Odds method (Hardt 2016) is designed to perfectly satisfy EOd notion. why this method can't satisfy EOd in your experiments?
> >
> > 3. Why GSTAR(DP) has smaller EOd as compared to GSTAR(EOd)?  GSTAR(EOd) has equalized odds fairness constraint while GSTAR(DP) considers demographic parity as a fairness constraint.
> >
> > 4. In general, if we care  EOd fairness notion, why we should use GSTAR(DP) or GSTAR(DP+EOd).
> >
> > And the last question:
> >
> > What is the advantage of your algorithm as compared to in-processing methods like Agarwal et al. "A Reductions Approach to Fair Classification". Since in algorithm 1, you are using the training dataset, we can directly use the reduction approach to find a fair classifier.

---

> > > ### Author Response · Authors · 2021-08-20
> > > **Response to more questions**
> > >
> > > Dear reviewer,
> > >
> > > Thank you for providing us with an additional opportunity to answer your questions. Please find our answers as follows.
> > >
> > > **[1. Mathematical meaning of EOd = 0.1]**
> > >
> > > Here, EOd is measured as the average odds difference, which is
> > > $$\frac{1}{2} \sum_{y \in \{0, 1\}}\Big| P(\hat Y=1|Y=y, A=0) - P(\hat Y=1|Y=y, A=0) \Big|.$$
> > > We can interpret EOd equals to 0.1 as that the average difference of true positive rate and false positive rate between privileged and unprivileged group equals to 0.1.
> > >
> > > **[2. Experimental results of EOd (Hardt et al., 2016)]**
> > >
> > > It is true that (Hardt et al., 2016) can achieve perfect EOd in training set by finding the mixing rate.
> > > However, the method (Hardt et al., 2016) is evaluated on the test set as all comparing methods for a fair comparison.
> > > Similar results can be found in Figures 1 and 3 of the FACT paper (Kim et al., 2020).
> > >
> > > **[3. Interpretation of experimental results]**
> > >
> > > In COMPAS, we observe improvements in total fairness violation with multiple fairness constraints employed. We deduce this could happen due to: 1) generalization of the estimated distribution from training data to testing data; 2) difference in the training and testing data distributions.
> > > For the training set, we achieve better fairness violation on the model with a single constraint, compared to the multi-constrained version or other single-constrained versions. In the training set of COMPAS data, we have the results as in the table [link-1].
> > >
> > >
> > > **[4. Benefit of multi-fairness over single target]**
> > >
> > > When we are targeting specific fairness (e.g., EOd), we indeed do not have to care about other fairness notions (e.g., DP).
> > > However, each fairness notion has its pros and cons in representing discrimination between groups.
> > > Thus, the desired algorithmic fairness may vary due to regulations, law, or the needs of different stakeholders or practitioners.
> > > If we can only optimize an algorithm to a particular fairness notion, we will lose the flexibility to take one or more fairness notions into account. Consequently, we will not be able to embrace diverse scenarios.
> > > GSTAR, however, has the ability to adapt to a different number of fairness notions as stated in the motivation and introduction of our paper. The choice of fairness notions depend on regulations, law, or the needs of different stakeholders or practitioners.
> > >
> > >
> > > **[5. Comparison with Agarwal et al.]**
> > >
> > > The advantages of our approach lie in several folds.
> > >
> > > First, our approach is more efficient. as described in line 189-196 of the main paper, the overall time complexity of our algorithm is $O(n + T)$, which is linear to the number of samples $n$ and the iteration number $T$. Our algorithm only needs to go through the data once to estimate the logit distribution. If we have the estimated distribution of logits available, then the time complexity can be further reduced to $O(T)$, which is independent of the number of samples. However, Agarwal et al. require complexity that is quadratic to the number of samples.
> > >
> > > Second, Agarwal et al. integrate various hyperparameters such as $\nu$, and $B$ to be tuned. In contrast, the only hyperparameter in our approach is $\lambda$, which balances between fairness and accuracy. We can set $\lambda$ in our approach depending on how much to focus on fairness/accuracy in the classification.
> > >
> > > Third, our approach enables an efficient way to adapt an existing model to a fairness notion. According to Figure 4 of our main paper, GSTAR can efficiently post-process an existing classifier to further improve fairness (here we use EOd as an example) without retraining the classifier. All we need to is to find an optimal threshold via GSTAR. However, since different fairness notions require different objectives in the in-processing method (Agarwal et al.), the adaptation of the method to a new fairness notion requires training the classifier from scratch. In complex domains (e.g., image), training a model from scratch is demanding as it involves extremely many parameters. In contrast, our approach can easily add or remove a particular notion to find the optimal threshold $\theta$ for fair classification.
> > >
> > > Fourth, our approach can improve fairness in classification while protecting individual-level sensitive information. Since GSTAR only requires the estimated distribution of logits for each demographic group, we do not need the sensitive information for each sample to find the optimal threshold.
> > >
> > >
> > > **[Reference]**
> > >
> > > [link-1] https://drive.google.com/file/d/1Jl0W3vV7pbrVpgjQO1HH6vQKG_pS7b-h/view?usp=sharing

---

> > > > ### Comment · Reviewer_5UxG · 2021-08-25
> > > > **One more question**
> > > >
> > > > Thank you for your clarification. I have one more question.
> > > >
> > > > How does the accuracy of $h(X)$ affects the accuracy of $\hat{Y}$?
> > > > Under what condition, your algorithm archive general optimality?
> > > >  A post-processing method may not be able to achieve general optimality, and its accuracy depends on the base classifier.
> > > >
> > > > Could you please explain whether your method can achieve the general optimality or not? By general optimality I mean the accuracy that an in-processing method (e.g., reduction approach Agarwal et al.) can achieve.
> > > >
> > > > I would like the author provides a theorem similar to Theorem 5.6 (Near optimality) in (Hardt et al. 2016).

---

> > > > > ### Author Response · Authors · 2021-08-30
> > > > > **Author response for the feedback**
> > > > >
> > > > > Dear reviewer,
> > > > >
> > > > > We greatly appreciate that the reviewer pointed us to Theorem 5.6 in Hardt et al. 2016, which is very relevant to the results the reviewer wanted us to show regarding on how the accuracy of $h$ affects the accuracy of $\hat{Y}$. Following the proof of their Theorem 5.6, we provide the following near optimality results for our method.
> > > > >
> > > > > **Theorem 5**  _With a bounded loss function $\ell$ and a given estimated density function $f$, let_ $h \in (-\infty, \infty)$ _be the induced random variable from the density_ $f$ and $\hat{R}_h = \sigma(h)$, where $\sigma(\cdot)$ is the softmax function. _Then the equalized odds predictor_ $\hat{Y}_h$ _derived from_ $(\hat{R}_h, A)$ _using the method in our paper can achieve near optimality in the following sense:_
> > > > > $$\mathbb{E}[\ell(\hat{Y}_h, Y)] \leq \mathbb{E}[\ell(Y^*, Y)] + 2 d_K(\hat{R}_h, R^*)$$
> > > > > _Here, $Y$ is the true label, $Y^*$ is the optimal equalized odds predictor derived from the Bayes optimal regressor $R^*$ as given in Hardt et al. (2016), and_ $d_K(\hat{R}_h, R^*)$ _is the conditional Kolmogorov distance._
> > > > >
> > > > > Before we prove the theorem, we first state the results from Lemma 5.5 proved in Hardt et al. (2016), which will be used in our proof.
> > > > >
> > > > > **Lemma** [Restatement of Lemma 5.5 in Hardt et al. (2016)]
> > > > > _Let_ $R, R^\prime \in [0,1]$ _be two random variables in the same probability space as $A$ and $Y$. Then, for any point $p$ on a conditional ROC curve of $R$, there is a point $q$ on the corresponding ROC curve of_ $R^\prime$ _achieving the same threshold such that_
> > > > > $$|| p - q||_2 \leq \sqrt{2} d_K(R, R^\prime), $$
> > > > >
> > > > > _where_
> > > > > $$ d_K (R, R^\prime) = \max\limits_{a,y\in\lbrace 0,1\rbrace} \sup\limits_{t\in[0,1]} \left| Pr \lbrace R > t |A=a, Y=y \rbrace - Pr\lbrace R^\prime > t |A=a, Y=y\rbrace\right|. $$
> > > > >
> > > > > **Proof.**
> > > > > Similar to Hardt et al. (2016), we focus on proving this theorem for equalized odds. The case of equal opportunity is analogous.
> > > > > The optimal classifier $Y^*$ corresponds to a point $p^*$.
> > > > > Under the equalized odds condition, our algorithm essentially finds the intersection point, $q$, of the two conditional ROC curves of $\hat{R}_h$ for $a=0$ and $a=1$. Then directly applying the above lemma, we get that
> > > > > $$ || p^* - q ||_2 \leq \sqrt{2} d_K(R, R^\prime) .$$
> > > > > The rest follows the same argument as in Theorem 5.6 of Hardt et al. (2016). That is, by assumption on the loss function, there is a vector $v$ with $||v||_2 \leq \sqrt{2}$ such that $\mathbb{E}[\ell(\hat{Y}_h, Y)] = \langle v,q \rangle$ and $\mathbb{E}[\ell(Y^*, Y)] = \langle v, p^* \rangle$. Applying Cauchy-Schwarz, we get
> > > > > $$ \mathbb{E}[\ell(\hat{Y}_h, Y)] - \mathbb{E}[\ell(Y^*, Y)] = \langle v, q-p^* \rangle
> > > > > \leq ||v||_2 \cdot || p^* - q ||_2 \leq 2 d_K(R, R^\prime). $$
> > > > >
> > > > >
> > > > > **Remark.**
> > > > > In Hardt et al., the point $q$ from their algorithm under equalized odds condition is the intersection point between two _line segments_, not the two ROC curves as in our paper. That is, assume without loss of generality that the first coordinate of $q_1$ (for group $a=1$) is greater than the first coordinate of $q_0$ (for group $a=0$) on the ROC curve plane; and that all points $p^*, q_0, q_1$ lie above the main diagonal. Then $q \in L_0 \cap L_1$ from their algorithm, where $L_0$ is the line segment between $q_0$ and the point $(1,1)$, and $L_1$ is the line segment between the point $(0,0)$ and $q_1$. As a result, in proving their Theorem 5.6, they need to show that $q$ from this construction satisfies $|| p^* - q ||_2 \leq 2 d_K(R, R^\prime).$ However, because the point $q$ from our algorithm lies on the ROC curve, we can directly apply the results from the lemma. This difference is further illustrated in the figure in [link-1] below, where the purple pentagram corresponds to $q$ found by our algorithm, and the green cross corresponds to $q$ from their algorithm. The figure shows the intersection points found from our algorithm versus Hardt et al.
> > > > >
> > > > > **[Reference]**
> > > > >
> > > > > [link-1] https://drive.google.com/file/d/1G5rKBGbpic1C7gD0d_BJRjYafHoPJaP2/view?usp=sharing

---

> > > > > > ### Comment · Reviewer_5UxG · 2021-09-04
> > > > > > **Thank you for your reponse**
> > > > > >
> > > > > >
> > > > > >
> > > > > > The theorem seems similar to theorem 5.6 of hard et al. However, in your method, in addition to the accuracy of the base classifier h(X), I believe that the Taylor approximation should affect the accuracy of the final predictor as well. I feel that somewhere in your proof, you have assumed that the Taylor approximation does not affect the accuracy of the final predictor. Could you please elaborate more on this theorem?
> > > > > >
> > > > > > Thank you.

---

> > > > > > > ### Author Response · Authors · 2021-09-05
> > > > > > > **Thank you for the follow-up question**
> > > > > > >
> > > > > > > Thank you for the follow-up question. We appreciate the additional opportunity for clarification.
> > > > > > >
> > > > > > > We would like to clarify that our algorithm connects to the classic family of Gauss-Newton algorithm to solve Nonlinear Least Squares Problem (NLSP) [1], and that the Taylor expansion is one step in this algorithm. The algorithm is an iterative algorithm, and the linearization via Taylor expansion is used in each step to iteratively solve the NLSP. The Gauss-Newton algorithm is guaranteed to converge to the optimal solution under certain technical conditions. Below we specify how our problem relates to the NLSP and conditions for the convergence. Given the convergence guarantee, we assume that optimal solution $\hat{Y}_h$ from this NLSP, the equalized odds predictor in Theorem 5 in our previous response, can be achieved via our algorithm and thus the error analysis in Theorem 5 in our previous response only contains the error from the base classifier $h(X)$. We hope this clarifies your question.
> > > > > > >
> > > > > > >
> > > > > > > #### **1) Connection between our problem and NLSP:**
> > > > > > > NLSP is to solve $$\min_{\theta} ||r(\theta)||^2_2,$$
> > > > > > > where $\theta$ is the decision variables and the objective function $r$ is a real vector function of $\theta$.
> > > > > > >
> > > > > > > In our case, the decision variable is the two-dimensional vector $\theta=(\theta_0, \theta_1)$ for $a=0$ and $a=1$ groups, and our objective function is the following 2-dimensional real vector function: $$r(\theta) = \big( r_1(\theta), r_2(\theta) \big)$$ with $$r_1(\theta) = \sqrt{L_{per}(\theta)},  r_2(\theta)  = \sqrt{\lambda L_{fair}(\theta)},$$ for $L_{per}$ and $L_{fair}$ in Equation (3) and (4) in the main paper. The $L_2$ norm $||r(\theta)||^2_2 = r_1(\theta)^2 + r_2(\theta)^2$ recovers the objective function in Equation (2) in the main paper. Recall that $\lambda$ plays the role of the Lagrangian multiplier for the fairness constraint. Under the optimal Lagrangian multiplier, $L_{fair}=0$, i.e., the perfect fairness condition is satisfied. For example,  the "GSTAR optimal" point illustrated in the Figure in [link-1] ([link-1] shown below in References) satisfies the perfect EOd condition. With the convergence guarantee specified below, our algorithm finds this point that has the near optimality property according to Theorem 5 given in our previous response.
> > > > > > >
> > > > > > >
> > > > > > > #### **2) Gauss-Newton algorithm:**
> > > > > > > The Taylor expansion converts the nonlinear optimization problem to a linear least square problem. That is, the parameter values are updated iteratively with $$\theta_{j} \approx \theta_{j}^{k+1} = \theta_{j}^{k} +\Delta_j,$$ at $k$-th iteration, with the vector of increments $\Delta=\lbrace\Delta_j\rbrace$ (a.k.a the shift vector). We linearize each component in $f$ to a first-order Taylor polynomial expansion as $$r_i (\theta) \approx r_i(\theta^k) + \sum_{j} \frac{\partial r_i(\theta)}{\partial\theta_j} \Delta_j.$$
> > > > > > > Plugging this equation into the objective function, we get the usual least square problem. Then, the optimal solution can be obtained as  $$\Delta = - (J^T J)^{-1} J^T f(\theta^k),$$ where $J=\{J_{ij}\}$ with $J_{ij} = \{\frac{\partial r_i(\theta)}{\partial\theta_j}\}$ is the Jacobian. Note that in GSTAR algorithm, we ignore the terms for $j\neq i$ in the Taylor expansion. Thus, we only kept the diagonal terms in $J^T J$ as
> > > > > > > $$\left(\frac{\partial  r_1(\theta)}{\partial\theta_j}\right)^2 + \left(\frac{\partial  r_2(\theta)}{\partial\theta_j}\right)^2$$
> > > > > > > for $j=0,1$. Plugging in the form of $r_1$ and $r_2$ as above, we achieve the solution provided in Equation (10) in the main paper.
> > > > > > >
> > > > > > >
> > > > > > > #### **3) Convergence of the Gauss-Newton algorithm:**
> > > > > > > The sequence of Gauss-Newton iterates {$\theta^k$} converges to $\theta^*$. Below we state the theory [1] on the sufficient conditions for convergence.
> > > > > > >
> > > > > > >
> > > > > > > The following assumptions are made in order to establish the theory.
> > > > > > >
> > > > > > > * A1. There exists $\theta^*$ s.t. $J^T(\theta^*)r(\theta^*) = 0$.
> > > > > > > * A2. The Jacobian at $\theta^*$ has full rank.
> > > > > > >
> > > > > > >
> > > > > > > We state Theorem 4 from [1] as follows:
> > > > > > >
> > > > > > > **Theorem 4 [1]** Assume that the estimated density function $f(\cdot)$ satisfy assumptions above. Further, $f(\cdot)$ satisfies that $$||Q(\theta^k)(J^T J)^{-1}(\theta^k)||_2 \leq \eta $$ for some constant $\eta\in[0,1)$ for each iteration $k$, where $Q(\theta)$ denotes the second order terms $\sum_i r_i(\theta) \nabla^2 r_i(\theta)$. Then as long as the initial solution is sufficiently close to the true optimal with $||\theta^0 -\theta^*||_2 \leq \epsilon$, the sequence of Gauss-Newton iterates {$\theta^k$} converges to $\theta^*$.
> > > > > > >
> > > > > > >
> > > > > > > **References**
> > > > > > >
> > > > > > > [1] Gratton, Serge, Amos S. Lawless, and Nancy K. Nichols. ``Approximate Gauss–Newton methods for nonlinear least squares problems.'' SIAM Journal on Optimization 18.1 (2007): 106-132.
> > > > > > >
> > > > > > > [link-1] https://drive.google.com/file/d/1G5rKBGbpic1C7gD0d_BJRjYafHoPJaP2/view?usp=sharing

---

> > > > > > > > ### Comment · Area_Chair_bCrR · 2021-09-10
> > > > > > > > **Follow-up comment on Theorem 5**
> > > > > > > >
> > > > > > > > Dear Authors,
> > > > > > > >
> > > > > > > > Thanks for the detailed response to each of our questions. I have a follow-up comment about Theorem 5 that you stated in response to Reviewer 5UxG.
> > > > > > > >
> > > > > > > > The result in Hardt et al. defines their measure of "closeness" in terms of conditional probability estimates (prediction | A, Y), whereas your paper (if I understand correctly) requires the conditional densities $h(X)$ to be close, which seems to be a much stronger requirement. It's not clear from your stated result how the two relate to each other: your theorem statement works with RVs $\hat{R}_h$ induced from the density $h$, but does this mean that your method does not necessarily require the estimated and true probability densities h(X) to be close, but only the induced RVs $\hat{R}_h$ to satisfy some notion of closeness? Do we expect  $d_K(\hat{R}_h, R^*)$ to be small for standard density estimation procedures?
> > > > > > > >
> > > > > > > > -AC

---

> > > > > > > > > ### Author Response · Authors · 2021-09-14
> > > > > > > > > **Thank you for the follow-up comment**
> > > > > > > > >
> > > > > > > > > Dear AC,
> > > > > > > > >
> > > > > > > > > Thank you for the follow-up comment. We appreciate the opportunity to provide clarification.
> > > > > > > > >
> > > > > > > > > We would like to clarify that the requirement for achieving the near optimality in our method (our Theorem 5) and in Hardt et al. (their Theorem 5.6) is the same. That is, the closeness between the conditional densities is required, not just the conditional probability estimates.
> > > > > > > > >
> > > > > > > > >
> > > > > > > > > To specify, the closeness requirement in Hardt et al. based on conditional Kolmogorov distance is shown in Equation (5.1) of their paper:
> > > > > > > > > $$
> > > > > > > > > d_K(\hat R,R') {\stackrel{\text{def}}{=}} \max\limits_{a,y\in\lbrace0,1\rbrace} \sup\limits_{t\in[0,1]} \left| Pr \lbrace R>t|A=a, Y=y \rbrace - Pr \lbrace R'>t|A=a, Y=y\rbrace \right|,
> > > > > > > > > $$
> > > > > > > > > where $R\in[0,1]$ and $R'\in[0,1]$ are real-valued scores, i.e., two regressors. Note that the distance is taking $sup$ over all $t\in[0,1]$, so this condition requires the entire conditional density curves from $R$ and $R^\prime$ to be close, not just close at some given threshold $t$.
> > > > > > > > >
> > > > > > > > > Next, the near optimality of Hardt et al. (their Theorem 5.6) shows:
> > > > > > > > > $$
> > > > > > > > > \mathbb{E}[\ell(\hat{Y}_h, Y)]
> > > > > > > > > \leq \mathbb{E}[\ell(Y^*, Y)] + 2 \sqrt{2}d_K(\hat{R}, R^*),
> > > > > > > > > $$
> > > > > > > > > where $R^*\in[0,1]$ is the Bayes optimal regressor and $\hat{R}\in[0,1]$ is a regressor whose density is estimated. In fact, the distribution function of their $\hat{R}$ corresponds to the score function $\sigma(h)$ in our paper, where $h$ is the logit and $\sigma(\cdot)$ is the softmax function.
> > > > > > > > >
> > > > > > > > >
> > > > > > > > > Seeing this connection, we stress that the closeness requirement in our result is the same as in Hardt et al., and that the near optimality of our algorithm (our Theorem 5) follows:
> > > > > > > > > $$
> > > > > > > > > \mathbb{E}[\ell(\hat{Y}_h, Y)]
> > > > > > > > > \leq \mathbb{E}[\ell(Y^*, Y)] + 2 d_K(\hat{R}_h, R^*),
> > > > > > > > > $$
> > > > > > > > > where $R^*\in[0,1]$ is the Bayes optimal regressor as given in Hardt et al., and $\hat{R}_h \in [0,1]$ comes from our estimated density, i.e., the distribution of $\hat{R}_h$ comes from by applying softmax function $\sigma(\cdot)$ on logit $h$.
> > > > > > > > >
> > > > > > > > > It is also worth mentioning that our method can obtain a better upper bound in near optimality, because in our method, the point $q$ under the equalized odds condition is the intersection point between the two ROC curves. While in Hardt et al., the point $q$ under the equalized odds condition is the intersection point between two **line segments**. Hence, we can directly apply the results from Lemma 5.5 in Hardt et al, while they need to derive a looser upper bound for their line intersection points using Lemma 5.5. Please also refer to the figure in [link-1] below for the difference, where the purple pentagram corresponds to $q$ found by our algorithm, and the green cross corresponds to $q$ from their algorithm. The figure shows the difference between the intersection points found from our algorithm versus Hardt et al.
> > > > > > > > >
> > > > > > > > >
> > > > > > > > > We hope this clarifies your question. Thank you again for your time and effort in reviewing our paper.
> > > > > > > > >
> > > > > > > > > [link-1] https://drive.google.com/file/d/1G5rKBGbpic1C7gD0d_BJRjYafHoPJaP2/view?usp=sharing

---

### Official Review · Reviewer_aSYp · 2021-07-16

**Rating:** 5
**Confidence:** 4

**Summary:**

The paper proposes a post-processing approach to achieve different notions of fairness. To do this, the paper first approximates the probability distribution of the output model. Group-specific confusion matrices can be derived by thresholding this distribution. The entries of such a matrix can be used to compute accuracy and various fairness metrics. The authors experimentally evaluate their approach on several datasets.

**Limitations And Societal Impact:**

I found the discussion on the limitation and societal impact to be satisfactory.

**Main Review:**

There are several concerns I have about this paper.

--First of all, while it is known that to achieve fairness optimal thresholding of the risk scores/output probabilities of the model should be group-specific, there are many instances in practice where the use of the sensitive attribute is forbidden. So it is not clear to me that the model can be used in practical applications.

--Connecting to the previous point, it is not clear where the benefit of the current approach compared to previous approaches is coming from. Is it because of a more clear technical approach or is it due to allowing group-specific thresholds (which would obviously perform better than approaches that utilize a single threshold). So in that respect, in the response, I would like to see the same curves being generated with only one threshold for all groups.

--The approach relies on utilizing models that output probabilistic decisions as opposed to deterministic ones. This certainly does not fall into the category of utilizing black-box models. So I found the use of the term "black-box" misleading.

--The approach also relies on good estimates of the distribution over probabilistic decisions of the model. According to the authors, the approach won't work when this distribution is mainly concentrated on the extremes. Hence, the approach is not suitable for truly black-box models. Moreover, the approach relies on a good estimation of this distribution. The authors propose several parametric families of distributions and choose the parameter which maximizes the likelihood. Is there a justification for why these families of distributions are sufficient? How does the error in estimating the probability distribution propagate to the result?

--By looking at the experimental results, it is not clear that the approach provides a better Pareto frontier than FACT. The more interesting regime in practice is where the fairness violation is at most 1%. In this regime, it seems like FACT consistently outperforms the current approach. Though for fairness violations that are much smaller (like 0.1% and lower), it seems the proposed approach is doing better. But how often in practice do we care about such small values of fairness violations?

--The optimization problem is not solved exactly. Are there any approximation guarantees?

--Many details of the experiments are missing. For example, what lambda values are used to derive the Pareto frontiers? Why there is only one accuracy/balanced accuracy are reported for different notions of fairness in Figure 3? Is this averaged over different notions of fairness?

Minor comments:

--The use of the log scale for the x-axis is a bit misleading in the figures. Is it because the approach only outperforms FACT in really small fairness violation values?

--The figures are very crowded. I have to zoom in to read the legend.

--It seems a bit suspicious that adding multiple notions of fairness, can improve the total fairness violation compared to using only one fairness metric in Figure 3. Is there any justification for this observation?

-------------------------------
Post-rebuttal: please see my official comment



**Time Spent Reviewing:**

5 hours

---

> ### Author Response · Authors · 2021-08-10
> **Author respond to answer the questions by the reviewer.**
>
> We thank the reviewer for taking the time to review our paper and provide constructive comments.
>
> **[Practicality of GSTAR]**
> Sensitive information is indeed hard to leverage in practice. Unlike other works that require sensitive information for each instance, we only require the distribution of logits for each sensitive group. Thus, GSTAR can be applied to relaxed scenarios where practitioners cannot access individual-level sensitive information but have estimated distributions of logits for each sensitive group. GSTAR utilizes both group-specific thresholds and estimates of the logit distribution to achieve performance and efficiency respectively. Note that GSTAR can learn one (unified) adaptive threshold for all sensitive groups when stronger privacy required.
>
> **[GSTAR with single threshold]**
> We show new experiments on COMPAS dataset to evaluate GSTAR with a single adaptive thresohld. Figure in [l-1] (please find all links in the reference below) presents the trend of fairness-accuracy tradeoff of two versions of GSTAR based on $\lambda$ value. Comparing with the baseline ($\theta = 0$), we observe that even with a single threshold in GSTAR (1 $\theta$ in the legend), the adaptive threshold helps to improve the fairness with comparable accuracy. However the improvement is not as significant as that of the group-wise version because it is impossible to achieve perfect fairness with a single threshold as the intersection of $f_{1a}$ and $f_{0a}$ differs by $a$. Figure in [l-2] shows the trend of learned $\theta$ based on $\lambda$ value. We see a single threshold version (black) lies between two thresholds of group-aware GSTAR in most cases. This implies that the single threshold converges to some point that gives up some of the fairness.
>
> **[Misleading terminology]**
> We thank the reviewer to give us an opportunity for clarification. By post-processing the "black-box model", we mean that we can improve the fairness of the model without knowing the model structure or parameters. Instead, we only need the output of the model. Our method focuses on learning an adaptive threshold to improve fairness in a model-agnostic manner, rather than utilizing the black-box model. We will clarify in final paper to avoid confusion.
>
> **[Quality of estimated distribution]**
> The performance of GSTAR indeed relies on the estimated distribution. We empirically found that the distribution of logits resembles some parametric distributions. Thus, we estimate the distribution with generally used parametric distributions such as Student's t-distribution by measuring the negative log-likelihood (NLL) in the training data. Note that GSTAR can be extended to a wide range of other distributions, even non-parametric distributions.
>
> For further analysis, we add new experiments by sweeping the parameters of parametric distribution to see the effect of the estimation quality. In COMPAS dataset, the best estimate (i.e., smallest NLL) of group $(y=0, a=0)$ with Student's t-distribution has parameters of df = 2.235, loc = -0.567, scale = 0.756 based on scipy package. To generate variations [l-3] of distributions with varying estimation qualities, we add noise $\alpha \in [-0.1, 100]$ to the scale of this distribution.
>
> In figure [l-4], we illustrate the trend of NLL (black), fairness violation (the lower the better), and accuracy (the higher the better) with varying noise ($\alpha$, x-axis). The color of lines follows the main paper. Dashed lines indicate the quantity of baseline model ($\theta = 0$). From this, we observed that the accuracy is the most sensitive to the change of estimation quality, while fairness is relatively stable.
>
> We will add the corresponding figures in the final paper. However, we assume the estimation is reliable and the guarantee on the estimation reliability is beyond our focus of this paper.
>
> **[Comparison with FACT Pareto]**
> GSTAR provides a better Pareto frontier than FACT since: 1) FACT has much steeper degradation of performance from a certain point; 2) GSTAR can potentially achieve comparable performance with small fairness violation. First, it has obvious benefit that ours have smoother curve than FACT Pareto. Second, in Fig. 2, we observe that most of the methods achieve the best possible accuracy (i.e., close to boundary) given a violation level. Moreover, in some method such as Adult and German, baseline model achieves fairness-accuracy tradeoff near the inflection point of FACT Pareto. That is, FACT would require a steep sacrifice of performance to improve fairness of the baseline. However, GSTAR shows a greater margin to improve fairness without losing accuracy.
>
> **[Approximation guarantee]**
> Our optimizing approach belongs to the family of Gauss-Newton algorithm to solve Nonlinear Least Squares Problem (NLSP). NLSP is to solve $$\min_{\theta} ||r(\theta)||^2_2,$$ where $\theta$ is the decision variables and the objective function $r$ is a real vector function of $\theta$. In our case, the decision variable is the two-dimensional vector $\theta=(\theta_0, \theta_1)$ for group 0 and group 1, and our objective function is the following 2-dimensional real vector function: $$r(\theta) = \big( r_1(\theta), r_2(\theta) \big)$$ with $$r_1(\theta) = \sqrt{L_{per}(\theta)},  r_2(\theta)  = \sqrt{\lambda L_{fair}(\theta)},$$ for $L_{per}$ and $L_{fair}$ in Equation (3) and (4) in the main paper. The $L_2$ norm $||r(\theta)||^2_2 = r_1(\theta)^2 + r_2(\theta)^2$ recovers the objective function in Equation (2).
>
> To solve NLSP, A classic family of Gauss-Newton Method is used. It converts the nonlinear optimization problem to a linear least square problem using Taylor expansion. That is, the parameter values are updated iteratively with $$\theta_{j} \approx \theta_{j}^{k+1} = \theta_{j}^{k} +\Delta_j,$$ at $k$-th iteration, with the vector of increments $\Delta=\lbrace\Delta_j\rbrace$ (a.k.a the shift vector). We linearize each component in $f$ to a first-order Taylor polynomial expansion as $$r_i (\theta) \approx r_i(\theta^k) + \sum_{j} \frac{\partial r_i(\theta)}{\partial\theta_j} \Delta_j.$$
> Plugging this equation into the objective function, we get the usual least square problem. Then, the optimal solution can be obtained as  $$\Delta = - (J^T J)^{-1} J^T f(\theta^k),$$ where $J=\{J_{ij}\}$ with $J_{ij} = \{\frac{\partial r_i(\theta)}{\partial\theta_j}\}$ is the Jacobian. Note that in GSTAR algorithm, we ignore the terms for $j\neq i$ in the Taylor expansion. Thus, we only kept the diagonal terms in $J^T J$ as
> $$\left(\frac{\partial  r_1(\theta)}{\partial\theta_j}\right)^2 + \left(\frac{\partial  r_2(\theta)}{\partial\theta_j}\right)^2$$
> for $j=0,1$. Plugging in the form of $r_1$ and $r_2$ as above, we achieve the solution provided in Equation (10) in the main paper.
>
> Studying the approximations to the nonlinear problems in Gauss–Newton algorithm and convergence property has a long history. The convergence of the algorithm is generally not guaranteed, and depends on the density estimation $f(\cdot)$. The following assumptions are required to establish the theory [1] on sufficient conditions for the convergence of the algorithm.
>
> * A1. There exists $\theta^*$ s.t. $J^T(\theta^*)r(\theta^*) = 0$.
> * A2. The Jacobian at $\theta^*$ has full rank.
>
> We state Theorem 4 from [1] as:
>
> **Theorem 4 [1]** _Assume that the estimated density function $f(\cdot)$ satisfy assumptions above. Further, $f(\cdot)$ satisfies that_ $$||Q(\theta^k)(J^T J)^{-1}(\theta^k)||_2 \leq \eta $$
> _for some constant $\eta\in[0,1)$ for each iteration $k$, where $Q(\theta)$ denotes the second order terms $\sum_i r_i(\theta) \nabla^2 r_i(\theta)$. Then as long as the initial solution is sufficiently close to the true optimal with_ $||\theta^0 -\theta^*||_2 \leq \epsilon$, _the sequence of Gauss-Newton iterates $\{\theta^k\}$ converges to $\theta^*$._
>
> The above sufficient conditions that guarantee convergence do not necessarily hold for general function $f(\cdot)$. Thus, protection against divergence is essential. Thus, we adopt a commonly used shift-cutting method. That is, we to reduce the length of the shift vector $\Delta$ by a fraction $\eta$, then the update becomes $$ \theta_j^{k+1} = \theta_j^k + \eta \Delta_j. $$
>
> **[Experimental setup]**
> For experimental setup, all comparing methods apply EOd as the fair constraint, thus we compare them via EOd in Fig. 2. Both the Pareto frontier from GSTAR and FACT are derived based on EOd constraint for a fair comparison. We follow the setup in Section G.3 of the FACT paper [Kim et al., 2020] to report their method, which does not require $\lambda$. For GSTAR, we estimate $f_{ya}$ and optimize $\theta_a$ from the training data, and report evaluation results (with the $\theta_a$ learned from training data) on the testing data. We use the same $\lambda$ for multiple fairness constraints for simplicity, but $\lambda$ can be introduced individually.
>
> Fig. 2 illustrates Pareto frontiers with 5 points of different $\lambda$ values in $[10^{-2},10^7]$ with equal logspace. For comparing methods, we sweep hyperparameters (e.g, weights for each term in the objective function) to visualize Pareto frontiers. Fig. 3 takes $\lambda$ or hyperparameter values from the upper-right point of the Pareto frontiers in Fig. 2, which indicates the best tradeoff for each method. Fig. 3 presents the 5 runs with the setup chosen based on the Pareto frontier to show the consistency of the performance of each model.
>
> Thanks for your insightful comments and we will include all the materials delivered in this response in final paper.
>
> **[Reference]**
>
> [1] Gratton et al. "Approximate Gauss–Newton methods for nonlinear least squares problems.'' SIOPT 18.1 (2007): 106-132.
>
> [l-1] https://1drv.ms/b/s!AlCY8gQvkMJOhP4IvJomKdxvniy2Sw?e=ddyL3z
>
> [l-2] https://1drv.ms/u/s!AlCY8gQvkMJOhP4HXbg-y6TGMHfRxg?e=2aaVrf
>
> [l-3] https://1drv.ms/u/s!AlCY8gQvkMJOhP4GxWDpM9XzOHXZlA?e=gexhj9
>
> [l-4] https://1drv.ms/u/s!AlCY8gQvkMJOhP4FAdHwKvjTzIm8WQ?e=EKTsQ1

---

> > ### Comment · Reviewer_aSYp · 2021-08-25
> > **Re: Rebuttal**
> >
> > I really appreciate the authors for providing detailed responses as well as even running some experiments to answer the questions. I think these experimental results should be added to the paper. However, I think there are several concerns that remain for me. The first one is the assumption that the distribution over probabilistic decisions belongs to some parametric class. As you can see in the additional experiments, deviation from this assumption can change the performance drastically. The second and perhaps the more important one is that the paper does not provide many advantages over FACT and most of the improvement is for very very low fairness violations which might not be that realistic in practice. I am going to change my rating from 4 to 5.

---

> > > ### Author Response · Authors · 2021-08-30
> > > **Author response for the feedback**
> > >
> > > Dear reviewer,
> > >
> > > Thank you for your attention to our paper and for providing us with an additional opportunity to answer your questions. Please find our response in the following.
> > >
> > > **[1. Limitation of parametric distribution]**
> > >
> > > We would like to clarify that our method is not limited to parametric distributions. Our method can be easily extended to non-parametric distribution estimation using kernel density estimation (KDE), which is a well-known density estimation method. Using an appropriate number of bins with sufficient granularity but not too detailed, we can construct a histogram.
> > > Then we do the sum of the Gaussian distributions with the mean as the center of the bins weighted by the number of samples in each bin to estimate the real distribution.
> > > Since the output of KDE is a mixture of Gaussians (i.e., Gaussian Mixture Model (GMM)), we can easily get the derivatives by adding the derivatives of each Gaussian model. After obtaining the density estimation and its derivative, the following procedure is the same as what is elaborated in our paper.
> > >
> > > Therefore, it is up to a practitioner who has domain knowledge to choose which method to take (i.e., parametric or non-parametric) and subsequent decision-making (e.g., which parametric distribution to use for parametric estimation; or how many bins to use for non-parametric estimation).
> > >
> > > It is also worth mentioning that we performed a numerical analysis of the influence of the estimated distribution quality in the previous response. We further provide a theoretical analysis for this, regarding the near optimality of our method. Please see our latest response to Reviewer 5UxG. We hope the theoretical and numerical analysis can help further clarify the property of our method.
> > >
> > > **[2. Trivial improvement comparing with FACT]**
> > >
> > > At a first glance, in some cases the improvement of GSTAR compared with FACT might seem trivial. However, we would like to clarify from the following two aspects.
> > >
> > > First, for the COMPAS and German data sets in Figure 2, the improvement of GSTAR is significant compared with FACT. For a fair comparison, we take GSTAR(EOd) as an example as we employed EOd constraint for FACT.
> > > FACT achieves 0.11 (COMPAS), 0.61 (German) of EOd violation, while GSTAR achieves 0.02 (COMPAS), 0.02 (German) at a similar accuracy level (similar y value in the figure).
> > > In other words, our GSTAR method improved the fairness violation by *82\%* and *97\%* on COMPAS and German data sets respectively, which is significant.
> > >
> > > Second, for CelebA and Adult data sets, it might seem less impactful comparing with the data sets above.
> > > FACT achieves an EOd violation of 0.06 in the CelebA and Adult data sets, while GSTAR achieves 0.04 at a similar accuracy level. This means *33.3\%* improvement in fairness.
> > > Although the ratio of improvement is large, we understand that the reviewer may have concern that the actual improvement value is relatively small.
> > > We would like to point out that in the well-known benchmark data sets, the performance improvement might seem minor, but the approach and efficiency of the novel method are still well recognized.
> > > A similar situation occurs in image classification in benchmark data sets such as ImageNet or CIFAR-10, which has an improvement within 1\% range [1].
> > > As described in the paper, ours is more efficient than other baselines with better fairness violations at the similar accuracy level.
> > > Our method even protects individual level sensitive information as we only require a group-wise density function to optimize the thresholds.
> > >
> > > More importantly, compared with other baselines including FACT, our GSTAR method achieves *consistently* best fairness, which is worth noting (and FACT is not always better than other methods). Consistency of a method is important for applying a method in practice, and is crucial in the field of fairness and equity.
> > >
> > > Thank you so much again for taking the time to review our paper and raising thoughtful questions. We will add the new results from theory and experiments to our final paper.
> > >
> > >
> > > **[Reference]**
> > >
> > > [1] Khan, Asifullah, et al. "A survey of the recent architectures of deep convolutional neural networks." Artificial Intelligence Review 53.8 (2020): 5455-5516.

---

### Author Response · Authors · 2021-08-24
**We are happy to answer more questions if there still exist concerns for our paper.**

Dear Reviewers,

Thanks for your time and efforts in reviewing our paper. We appreciate your constructive comments. Hopefully, our response can address your concerns.

If you have further questions or confusion, we would be very happy to clarify. Thank you very much.

Best,

Authors

---

### Decision · Program_Chairs · 2021-09-27

**Decision:**

Reject

**Comment:**

The paper proposes a post-processing technique to tune group-specific thresholds on a classifier to satisfy fairness constraints. The consensus from a majority of the reviewers is that the paper needs more work and is not ready for publication in its current form.

One of the main concerns is the assumption that the "estimated probability density function" follows a parametric form, and that it's unclear how practical this assumption is, and what happens when it fails or only holds approximately. A second concern is that the proposed threshold-tuning procedure does not come with a convergence guarantee. In response, the authors provide two separate results: (i) an asymptotic convergence guarantee for their optimization algorithm under some initial conditions, (ii) an adaptation of a result from Hardt et al. (2016) to show that the fairness constraint is approximately satisfied under some conditions on the estimated probability density.

Both these are however substantial additions to the paper, and would require another round of review. Hence we strongly encourage the authors to use the feedback provided to improve their manuscript and submit it to a future venue.

P.S. I appreciate the authors clarification on the similarities between the requirements in their paper and that in Hardt et al. (and the potential advantage that their approach offers in providing a tighter bound), but would have to give it some more thought. Bringing out some of these differences/similarities with Hardt et al. would make the paper more compelling.